

# The Effect of Meteorological Conditions and Atmospheric Composition in the Occurrence and Development of New Particle Formation (NPF) Events in Europe

**Dimitrios Bousiotis[1], James Brean[1], Francis Pope[1] , Manuel Dall'Osto[2]**
**Xavier Querol[3], Andres Alastuey[3], Noemi Perez[3], Tuukka Petäjä[4]**
**Andreas Massling[5], Jacøb Klenø Nøjgaard[5], Claus Nørdstrom[5]**
**Giorgos Kouvarakis[6], Stergios Vratolis[7], Konstantinos Eleftheriadis[7]**
**Jarkko V. Niemi[8], Harri Portin[8] and Roy M. Harrison[1a*]**

**[1]Division of Environmental Health and Risk Management**
**School of Geography, Earth and Environmental Sciences**
**University of Birmingham, Edgbaston, Birmingham B15 2TT, United Kingdom**

**[2]Institute of Marine Sciences, Passeig Marítim de la Barceloneta, 37-49 E-08003**
**Barcelona, Spain**

**[3]Institute of Environmental Assessment and Water Research (IDAEA - CSIC)**
**08034, Barcelona, Spain**

**[4]Institute for Atmospheric and Earth System Research (INAR) / Physics, Faculty of Science**
**University of Helsinki, Finland**

**[5]Department for Environmental Science, Aarhus University, DK-400, Roskilde, Denmark**

**[6]Environmental Chemical Processes Laboratory (ECPL), Department of Chemistry,**
**University of Crete, 70013, Heraklion, Greece**

**[7]Environmental Radioactivity Laboratory, Institute of Nuclear and Radiological Science &**
**Technology, Energy & Safety, NCSR Demokritos, Athens, Greece**

**[8]Helsinki Region Environmental Services Authority (HSY),**
**FI-00066 HSY, Helsinki, Finland**

**[a]Also at: Department of Environmental Sciences / Center of Excellence in Environmental**
**Studies, King Abdulaziz University, PO Box 80203, Jeddah, 21589, Saudi Arabia**





**ABSTRACT**
Although new particle formation (NPF) events have been studied extensively for some decades, the
mechanisms that drive their occurrence and development are yet to be fully elucidated. Laboratory
studies have done much to elucidate the molecular processes involved in nucleation, but this
knowledge has yet to be linked to NPF events in the atmosphere, except at very clean air sites.
There is great difficulty in successful application of the results from laboratory studies to real
atmospheric conditions, due to the diversity of atmospheric conditions and observations found, as
NPF events occur almost everywhere in the world without following a clearly defined trend of
frequency, seasonality, atmospheric conditions or event development. The present study seeks
common features in nucleation events by applying a binned linear regression over an extensive
dataset from 16 sites of various types (rural and urban backgrounds as well as roadsides) in Europe.
A clear positive relation is found between the solar radiation intensity, temperature and atmospheric
pressure with the frequency of NPF events, while relative humidity presents a negative relation with
NPF event frequency. Wind speed presents a less consistent relationship which appears to be
heavily affected by local conditions. While some meteorological variables appear to have a crucial
effect on the occurrence and characteristics of NPF events, especially at rural sites, it appears that
their role becomes less marked when at higher values.
The analysis of chemical composition data presents interesting results. Concentrations of almost all
chemical compounds studied (apart from $O_3$) and the Condensation Sink (CS) have a negative





relation with NPF event probability, though areas with higher average concentrations of $SO_2$ had
higher NPF event probability. Particulate Organic Carbon (OC), Volatile Organic Compounds
(VOCs) and particulate phase sulphate consistently had a positive relation with the growth rate of
the newly formed particles. As with some meteorological variables, it appears that at increased
concentrations of pollutants or the CS, their influence upon NPF probability is reduced.





# 1.   INTRODUCTION

New Particle Formation (NPF) events are an important source of particles in the atmosphere
(Merikanto et al., 2009; Spracklen et al., 2010), which are known to have adverse effects on human
health (Schwartz et al., 1996; Politis et al., 2008; Kim, et al., 2015) as well as affecting the optical
and physical properties of the atmosphere (Makkonen et al., 2012; Seinfeld and Pandis, 2012).
While they occur almost everywhere in the world (Dall'Osto et al., 2018; Kulmala et al., 2017;
O'Dowd et al., 2002; Wiedensohler et al., 2019; Chu et al., 2019; Kerminen et al., 2018), great
diversity is found in the atmospheric conditions within which they take place. Many studies have
been done in a large number of different types of locations (urban, traffic, regional background)
around the world and differences were found in both the seasonality and intensity of NPF events.
To an extent this variability is due to the mix of conditions that are specific to each location, which
blurs the general understanding of the conditions that are favourable for the occurrence of NPF
events (Berland et al., 2017; Bousiotis et al., 2020). For example, solar radiation is considered as
one of the most important factors in the occurrence of NPF events (Kulmala and Kerminen, 2008;
Kürten et al., 2016; Pikridas et al., 2015; Salma et al., 2011), as it is needed for the photochemical
reactions that lead to the formation of sulphuric acid (Petäjä et al., 2009; Cheung et al., 2013),
which is considered as the main component of the formation and growth of the initial clusters (Iida
et al., 2008; Weber et al., 1995); although in many cases, NPF events did not occur in the seasons
with the highest insolation (Park et al., 2015; Vratolis et al., 2019). Similarly, higher temperatures
are considered favourable for the growth of the newly formed particles as increased concentrations





of both Biogenic Volatile Organic Compounds (BVOCs) and Anthropogenic Volatile Organic
Compounds (AVOCs) (Yamada, 2013; Paasonen et al., 2013) and their oxidation products (Ehn et
al., 2014) are associated with the growth of the particles. This appears to be true in most cases, as
higher growth rates are found in most cases in the local summer (Nieminen et al., 2018), although
the actual importance of those VOCs in the occurrence of NPF events is still not fully elucidated.
The effect of other meteorological variables is even more complex, with studies presenting mixed
results on the effect of the wind speed and atmospheric pressure. Extreme values of those variables
may be favourable for the occurrence of NPF events, as they are associated with increased mixing
in the atmosphere, but at the same time suppress due to increased dilution of precursors (Brines et
al., 2015; Rimnácová et al., 2011; Shen et al., 2018; Siakavaras et al., 2016), or favour them due to
a reduced condensation sink (CS).

The effect of atmospheric composition on NPF events is also a puzzle of mixed results. While the
negative effect of the increased CS is widely accepted (Kalkavouras et al., 2017 ; Kerminen et al.,
2004; Wehner et al., 2007), cases are found when NPF events occur on days with higher CS
compared to average conditions (Größ et al., 2018; Kulmala et al., 2005). Sulphur dioxide ($SO_2$),
which is one of the most important contributors to many NPF pathways, in most studies was found
in lower concentrations on NPF event days compared to average conditions (Alam et al., 2003;
Bousiotis et al., 2019), although there are studies that have reported the opposite (Woo et al., 2001;
Charron et al., 2008). Additionally, in a combined study of NPF events in China, events were found



to be more probable under sulphur-rich conditions rather than sulphur-poor (Jayaratne et al., 2017).
Similar is the case with the BVOCs and AVOCs, which present great variability depending the area
studied (Dai et al., 2017), and their contribution in the growth of the particles is not fully understood
yet. Until recently, it was considered unlikely for NPF events as they are considered in the present
study, deriving from secondary formation not associated with traffic related processes such as
dilution of the exhaust, to occur within the complex urban environment due to the increased
presence of compounds, mainly associated with combustion processes, which would suppress the
survival of the newly formed particles within this type of environment (Kulmala et al., 2017).
Despite that though, NPF events were found to occur within even the most polluted areas and
sometimes with high formation and growth rates (Bousiotis et al., 2019; Yao et al., 2018).
It is evident that while a general knowledge of the role of the meteorological and atmospheric
variables has been achieved, there is great uncertainty over the extent and variability of their effect
(and for some of them even their actual effect) in the mechanisms of NPF in real atmospheric
conditions, especially in the more complex urban environment (Harrison, 2017). The present study,
using an extensive dataset from 16 sites in six European countries, attempts to elucidate the effect
of several meteorological and atmospheric variables not only in general, but also depending on the
geographical region or type of environment. While studies with multiple sites have been reported in
the past, to our knowledge this is the first study that focuses directly on the effect of these variables
upon the probability of NPF events as well as the formation and growth rates of newly formed
particles in real atmospheric conditions.



## 124  2.     DATA AND METHODS

### 125  2.1     Site Description and Data Availability

The present study uses a total of more than 85 years of hourly data from 16 sites from six countries
of Europe of various land usage and climates from which 1950 NPF events were extracted and
studied. A list of the available data and a brief description for each site is found in Table 1 (for the
ease of reading the sites are named by the country of the site followed by the last two letters which
refer to the type of site, being RU for rural/regional background, UB for urban background and RO
for roadside), while a map of the sites is found in Figure 1. The NPF frequency and formation rate
for each site is found in Table 2.

### 134  2.2     Methods

### 135  2.2.1   NPF events selection

NPF events were selected using the method proposed by Dal Maso et al (2005). As of this, an NPF
event is considered when a new mode of particles appears in the nucleation mode, prevails for some
hours and shows signs of growth. The events can then be classified into classes I and II according to
the level of confidence, while class I events can be further classified to Ia and Ib, with Ia events
having both a clear formation of a by new mode of particles as well as a distinct growth of the new
mode of particles, while Ib consists of rather clear events that fail though by at least one of the
criteria set. In the present study, only the events of class Ia were considered with the additional
criterion of at least 1 nm h$^{-1}$ growth for at least 3 hours.





### 144  2.2.2  Calculation of condensation sink, growth rate, formation rate, and NPF event

probability

The condensation sink (CS) is calculated according to the method proposed by Kulmala et al.,
(2001) as:

$CS = 4\pi D_{vap} \sum \beta_M \, r \, N$

where r and N is the radius and number concentration of the particles and $D_{vap}$ is the diffusion
coefficient calculated as (Poling et al., 2001):

$D_{vap} = 0.00143 \cdot T^{1.75} \dfrac{\sqrt{M_{air}^{-1} + M_{vap}^{-1}}}{P\left(D_{x,air}^{\frac{1}{3}} + D_{x,vap}^{\frac{1}{3}}\right)^2}$

for T = 293 K and P = 1013.25 mbar. M and $D_x$ are the molar mass and diffusion volume for air and
sulphuric acid. $\beta_M$ is the Fuchs correction factor calculated as (Fuchs and Sutugin, 1971):

$\beta_M = \dfrac{1 + K_n}{1 + \left(\dfrac{4}{3a} + 0.377\right)K_n + \dfrac{4}{3a}K_n^{\,2}}$





where $K_n$ is the Knudsen number, calculated as $K_n = 2\lambda_m/d_p$ where $\lambda_m$ is the mean free path of the
gas.

Growth rate (GR) is calculated as (Kulmala et al., 2012):

$$GR = \frac{D_{P_2} - D_{P_1}}{t_2 - t_1}$$

for the size range between the minimum available particle diameter up to 30 nm (50 nm for the UK
sites due to the higher minimum particle size available). The time window used for the calculation
of the growth rate was from the start of the event until a) growth stopped, b) GMD reached the
upper limit set or c) the day ended.

The formation rate J was calculated using the method proposed by (Kulmala et al., 2012) as:

$$J_{d_p} = \frac{dN_{d_p}}{dt} + CoagS_{d_p} \times N_{d_p} + \frac{GR}{\Delta d_p} \times N_{d_p} + S_{losses}$$

where $CoagS_{dp}$ is the coagulation rate of particles of diameter $d_p$, calculated as (Kerminen et al.,

2001):






$$\mathrm{CoagS}_{d_p} = \int K\big(d_p, d'_p\big)\, n\big(d'_p\big) dd'_p \;\cong\; \sum_{d'_p = d_p}^{d'_p = \max} K\big(d_p, d'_p\big)\, N_{d_p}$$

$K(d_p, d'_p)$ is the coagulation coefficient of particles with diameters $d_p$ and $d'_p$, while $S_{losses}$ accounts
for additional loss terms (i.e. chamber wall losses), which are not applicable in the present study.
For the present study, the formation rate of particles of diameter of 10 nm was calculated for
uniformity (16 nm for the UK sites), though most sites had data for particle sizes below 10 nm.

The NPF probability was calculated by the number of NPF event days divided by the number of
days with available data in the given group (temporal, wind direction sector etc.). The results
presented in this study were also normalised according to the data availability, as:

$$NPF_{probability} = \frac{N_{NPF\ event\ days\ for\ group\ of\ days\ X}}{N_{days\ with\ available\ data\ for\ group\ of\ days\ X}}$$

**2.2.3   Calculation of the slope and intercept for the variables used**
Due to the large datasets available and the great spread of the values, a direct comparison between a
given variable and any of the characteristics associated with NPF events (NPF probability, growth
rate and formation rate) always provided results with low significance. As a result, an alternative
method which can provide a reliable result without the noise of the large datasets was used in the



present study, to investigate the relations between the variables which are considered to be
associated with the NPF events. For this, a timeframe which is more directly associated with the
NPF events typically observed in the mid-latitudes was chosen. For NPF probability and GR the
timeframe between 05:00 to 17:00 LT was chosen, which is considered the time when the vast
majority of NPF events take place and further develop with the growth of the particles. For the
formation rate a smaller timeframe was chosen, 09:00 to 15:00 LT (Local Time) which is ± 3 hours
from the time of the maximum formation rate found for almost all sites (12:00 LT). This was done
to exclude as far as possible the effect of the morning rush at the roadsides, as well as only to
include the time window when the formation rate is mostly relevant to NPF events (negative values
that are more probable outside this timeframe would bias the results).

Specifically, for the CS the timeframe 05:00 to 10:00 LT was chosen. This was done to avoid
including the direct effect of the NPF events as well as to provide results for the conditions which
either promote or suppress the characteristics studied, which specifically for the CS are more
important before the start of the events. The extreme values (very high or very low) which bias the
results only carrying a very small piece (forming bins of very small size) of information were then
removed, though 90% of the available data were used for all the variables. The data left was
separated into smaller bins and a minimum of 10 bins was required for each variable (for example if
the difference between the minimum and the maximum relative humidity (RH) is 70%, then 14 bins



each with a range of 5% were formed). The variables of interest were then averaged for each bin
and plotted, and a linear relation was considered for each one of them.

The slope of the linear relations ($a_N$, $a_G$ and $a_J$ for NPF probability, growth rate and formation rate $J_{10}$
accordingly) found in this analysis should be used with great caution as apart from the atmospheric
conditions (local and meteorological as well as atmospheric composition) it is also affected by the
variable in question (e.g. a greater NPF probability will provide a greater slope), resulting in giving
the same trend for all the atmospheric variables tested; the sites with the higher values of these
variables (NPF probability and formation rate) always had greater slope values and vice versa.  In
order to remove the effect of the variable in question (NPF probability or formation rate – growth
rate will provide an untrustworthy result as it is calculated in a different range for each site due to
the lower available size of particles), the slopes were normalised by dividing them by their
respective variable (e.g. divide the slope of the NPF probability with the NPF probability),
providing with a new normalised slope ($a_N$* for NPF probability or $a_J$* for the formation rate) that
will have no significance other than its absolute value, which can be used for direct comparisons:
$$a_N^* = \frac{a_N}{\text{NPF \%}}$$

Where $a_N$ is the slope of the relation between the given variable and NPF probability (NPF %)

$$a_J^* = \frac{a_J}{J_{10}}$$





Where $a_J$ is the slope of the relation between the given variable and the formation rate of 10 nm
particles $J_{10}$ ($J_{16}$ for the UK sites).

**3.      RESULTS**
In this study NPF events are generally observed as particles grow from a smaller size (typically 3-
15 nm depending on the size detection limit of instruments used) to 30 nm or larger.  They therefore
reflect the result both of nucleation, which creates new particles of 1-2 nm (not detected with the
instruments used in this study), and growth to larger sizes. In analysing NPF events, we therefore
consider three diagnostic features:
•     the frequency of events occurring (i.e. days with an event divided by total days with relevant

data),

•     the rate of particle formation at a given size ($J_{10}$ in this case),
•     the growth rate of particles from the lower measurement limit to 30 nm (or 50 nm for the UK

sites).


**3.1      Meteorological Conditions**
The slopes and $R^2$ from the analysis of the meteorological variables, as well as the average
conditions of these variables are found in Table 3. The results for each site and variable are found in
Figure S1.





### 3.1.1 Solar radiation intensity


As mentioned earlier, solar radiation is considered as one of the most important variables in NPF
occurrence, as it contributes to the production of $H_2SO_4$ which is a main component of the initial
clusters and participates in the early growth of the newly formed particles. Hidy et al. (1994)
reported up to six times higher $SO_2$ oxidation rates into $H_2SO_4$ in typical summer conditions
compared to winter). For almost all sites this relation is confirmed with very strong correlations
between the intensity of solar radiation and the probability for NPF events. The relation between the
solar radiation and NPF probability was positive at all sites and only three sites (FINUB, SPARU
and GREUB) presented weak correlations ($R^2$ below 0.40). Weaker correlations were found for the
southern European sites, which might be associated with the higher averages for solar radiation, or
the interference of other processes (such as coinciding with increased CS by recirculation of air
masses (Carnerero et al., 2019), possibly making it less of an important factor for these areas.

The relationship of solar radiation to the growth rate was weaker in all cases and did not present a
clear trend. A few sites presented a strong correlation, which in all cases were background sites
(either rural or urban). The relation found in most cases was positive apart from two roadsides and
GREUB, though due to the low $R^2$ these results cannot be used with confidence. It seems though
that the solar radiation intensity is probably a more important factor at background sites rather than
at roadsides, where possibly local conditions (such as local emissions) are more important. Finally,
the formation rate has a positive relation with the solar radiation intensity, with strong correlations



in most areas. The correlations were stronger at the rural background sites compared to the
roadsides, which further underlines the increased importance of this factor at this type of site. A
negative relation between the solar radiation intensity and the formation rate was found at the
GRERU site but the $R^2$ is very low.

Plotting the normalised slopes for NPF event probability $a_N{}^*$ with the average solar radiation at each
site (Figure 2) a negative relation is found ($R^2 = 0.62$), with the southern areas (those with higher
average solar intensity) having smaller $a_N^*$ compared to those in higher latitudes (and thus with a
lower average solar radiation). This may indicate that while solar radiation is a deciding factor in
the occurrence of an NPF event, when in greater intensity its role becomes relatively less important,
a finding that was also implied by Wonaschütz et al. (2015). Additionally, the $a_J^*$ was found to be
higher at all rural sites compared to their respective roadsides (and urban background sites for all
but the Greek and German ones), making it a more important factor at this type of site (Figure 3).

### 3.1.2    Relative humidity

Relative humidity is considered to have a negative effect on the occurrence of NPF events (Jeong et
al., 2010; Hamed et al., 2011; Park et al., 2015; Dada et al., 2017; Li et al., 2019). While water in
the atmosphere is one of the main compounds needed for the formation of the initial clusters either
on the binary or ternary nucleation theory (Korhonen et al., 1999; Mirabel and Katz, 1974), in
atmospheric conditions it may also play a negative role suppressing the number concentrations of





new particles by increasing aerosol surface area. Consistent with this, a negative relation of the RH
with NPF probability was found for all the sites of this study with very high $R^2$ for almost all of
them. This is not simple to interpret as solar radiation, temperature, RH and CS are not independent
variables, since an increase in temperature of an air mass due to increased solar radiation will be
associated with reduced RH, which in turn affects the CS.  The sites in Greece presented lower $R^2$
compared to the other sites while, GRERU was found to have the weakest correlation. Growth rate
on the other hand had a variable relation, either positive or negative, with only a handful of
background sites having strong correlations. Among these the German background sites as well as
FINRU, which were among the sites with the highest average RH (average RH for GERRU is
81.9%, GERUB is 78.7% and FINUB is 80.1%) presented a negative relation between the RH and
growth rate, while DENRU (average RH at 75.7%) had a positive relation, which might indicate
that the relation between these two variables may vary depending upon the RH range. Formation
rate also appears to have a negative relation with the RH, though this relation was significant ($R^2 >$
0.40) for only 6 sites, which once again in most cases are sites with higher RH average conditions.
Along with the results of the growth rate this might indicate that the RH becomes a more important
factor in the development of NPF events as its values increase.

The normalised slopes once again provide some additional information. Regarding the NPF
probability, it is found that the $a_N$* was more negative at rural sites compared to roadsides. This
indicates that the RH has a smaller effect at roadsides, as other variables, such as the atmospheric





composition, are probably more important within the complex environment in this type of sites.
Additionally, the relation between $a_N$* and average RH at the sites had a negative relation ($R^2$ =
0.46), which further shows that the RH becomes a more important factor at higher values (Figure
4). Furthermore, at the sets of rural and roadside sites with $R^2$ higher than 0.40 for the relation
between RH and the formation rate (UK and German sites), it was found that the $a_J$* was more
negative at the rural sites which indicates that the RH is a more important factor at rural sites
compared to their respective roadsides.

### 3.1.3    Temperature

Temperature can have both a direct and indirect effect in the development of NPF events, as it is
directly associated with the abundance of biogenic volatile carbon which is an important group of
compounds whose oxidation products can participate in nucleation itself (Lehtipalo et al., 2018;
Rose et al., 2018), as well as in the growth of newly formed particles, while it may affect the
particle size distributions or number concentrations through other processes such as particle
evaporation. Most of the sites of the present study presented a strong relation of NPF probability
with temperature, which in most cases was positive, though in many cases (such as the Danish,
Spanish and Finnish sites) there seems to be a peak in the NPF probability at some temperature,
after which a decline starts (though being at the higher end does not greatly affect the results). Sites
with smaller $R^2$ (weaker association with temperature), were mainly those that have a seasonal
variation that favoured seasons other than summer. These sites not only had weaker relation of NPF


probability with temperature, but in most cases had a negative relation (background sites in Finland,
Spain and Greece). The Finnish sites, having the lowest average temperatures and a sufficient
amount of data below zero temperature, show at all three sites the possible presence of a peak in the
NPF event probability for temperatures below zero. This seems to be the cause of the weak relations
found there and they seem to be associated with the formation rate $J_{10}$, which also seems to have an
increasing trend below zero degrees.  This may be the result of increased stability of molecular
clusters at lower temperatures, as well as the possible enhancement of growth mechanisms in lower
temperatures (below 5°C) by other chemical compounds in the atmosphere (i.e. nitric acid and
ammonia) as found by Wang et al., (2020). Laboratory experiments show that the characteristics of
organic aerosol forming from alpha-pinene is governed by gas phase oxidation (e.g. Ye et al. 2019).
In the real atmosphere, the higher temperature enhances the amount of biogenic vapours (e.g.
Paasonen et al. 2013), and although the oxidation can be more efficient in higher temperatures, the
lower temperatures favour formation of more non-volatile compounds (Ye et al. 2019; Stolzenburg
et al. 2018).

Growth rate had a more uniform trend, with almost all sites having a positive relation with
temperature (apart from GERRO, though with $R^2 = 0.00$). This relation was very strong for most
sites, which is also confirming the summer peak found for the growth rate at most of these sites. A
strong relation with temperature was also found for the formation rate for most sites, and was
positive for almost all sites (apart from FINRO with $R^2 = 0.01$ and the Greek sites). As with the



NPF probability, in general the sites with a seasonal variation of events that favoured summer had
the strongest relation (high $R^2$) of the formation rate with temperature, which might indicate that
this variable, either through its direct or indirect effect is an important one for the seasonal
variability of NPF events in a given area.

The normalised slopes for this variable did not present a clear trend among the areas studied, other
than presenting greater $a_N*$ for the sites with a summer peak in their NPF event seasonal variation.
As with other meteorological variables, the importance of this variable became smaller with
increased values in the average conditions for both the NPF probability (Figure 5) and $J_{10}$, though
these relations were not significant (biased by the very low average temperatures and different
behaviour of the variables at the Finnish sites, without which the relation becomes a lot clearer as
pointed in Figure S2). The variation though within the sites of the same area (different sites in same
country / region) appears to directly follow the variability of temperature, showing that the
temperature directly affects the occurrence of NPF events when other factors remain constant,
having a negative trend for all countries but Finland. The $a_J{}^*$ though is found to be greater
(positively or negatively) at the rural background sites than at the other two types of sites at all areas
studied, showing that it is a more important factor for the formation rate at this type of site
compared to others (Figure 6).




### 3.1.4    Wind speed


Wind speed may have both a positive and a negative effect on the occurrence of NPF events. On
one hand, it may promote NPF events by the increased mixing of the condensable compounds in the
atmosphere as well as by reducing the CS, while on the other hand high wind speeds may suppress
NPF events due to increased dilution. It should be considered that the variability found is also
affected by the specific conditions found at each site. The wind speed measurements in many cases,
especially in urban sites, can be biased by the local topography or specific conditions found at each
site, thus representing the local conditions for this variable rather than the regional ones. Similarly,
measurements of wind speed at well sited meteorological stations may be more representative of
regional conditions, than of those affecting the sites of nucleation measurement.  The sites in this
study presented mixed results, both in the importance as well as the effect of the wind speed
variability. Three different behaviours were found in the variation of NPF event probability and
wind speed which appear to be associated with local conditions as they are almost uniformly found
among the sites within close proximity. Some sites presented a steady increase of NPF event
probability with wind speed (Danish sites as well as UKUB, FINRU, SPAUB and GRERU), while
others were found to steadily decline with increasing wind speeds (German sites – it should be
noted that the German sites are the only ones that are located at a great distance from the sea), while
some were found to reach a peak and then decline, which also leads to smaller $R^2$ (UKRU, UKRO,
SPARU and to a lesser extent GREUB). The reasons for these differences between the sites are very
hard to distinguish as apart from the wind speed the origin and the characteristics of these air



masses play a crucial role. Following this, it appears that NPF probability is very low or zero for
wind speeds close to calm for the sites with an increasing trend (as well as those that have a peak
and decline after), while the opposite is observed for the German sites where the maximum NPF
probability is found for very low wind speeds.

Similarly, the effect of different wind speeds upon the growth rate also varied a lot, though it was
found to be negative in all the cases where $R^2$ was higher than 0.50 (UKUB, DENRU, DENRO,
GERRU, GERUB and GREUB). Finally, the formation rate was found to have a significant
correlation only at two sites (UKRO and DENRU), probably indicating that the variability of the
wind speed either does not affect this variable or its effect is rather small.

The normalised slopes did not have any notable relation to either the NPF probability or the
formation rate further confirming that the effect of the different wind speeds is not due to its
variability only, but it is also influenced by the characteristics of the incoming air masses as well as
specific local conditions found at each site.

**3.1.5   Pressure**
In almost all the sites with available data (apart from the Spanish), the NPF probability presented a
positive relation with high significance at all types of sites. The greater significance found at the
rural sites indicates the increased importance of meteorological conditions in the occurrence of NPF



events at this type of site. The growth rate also presented a similar picture, with positive relations at
all the background sites of this study except the ones in Greece and FINUB (though with low $R^2$ at
0.02). This is probably associated with the seasonal variation found in Greece where higher growth
rates were found in summer, a period when increased wind speeds and lower atmospheric pressure
was found due to the Etesians (Kalkavouras et al., 2017). An interesting find is the negative slopes
found at all the roadsides, though the significance of these results is relatively low ($R^2 < 0.43$) and
always lower compared to the rural sites. The effects of pressure above are not likely to be
important.  Once again however, this is not an independent variable and higher pressure in summer
tends to be associated with higher insolation and temperatures and lower RH.  Since most events
occur in the warmer months of the year, this is probably the explanation for the apparent effects of
pressure. The formation rate presented relations of low significance for the sites of this study. Due
to this, pressure should not be an important factor for the formation rate at any type of site.

The normalised slopes did not present any clear trends, even for the NPF probability for which the
results presented significant relations at almost all sites.

**3.2       Atmospheric Composition**
The slopes and $R^2$ from the analysis of a number of air pollutants and the condensation sink, as well
as the average conditions of these variables are found in Table 4. The results for each site and
variable are found in Figure S1.





### 3.2.1     Sulphur dioxide (SO$_2$)

Sulphur dioxide is considered as one of the main components that participate in the NPF process.
According to nucleation theories and observations, H$_2$SO$_4$ is the most important compound from
which the initial clusters are formed, as well as one of the candidate compounds for the initial steps
of particle growth (Kirkby et al., 2011; Nieminen et al., 2010; Sipila et al., 2010). As H$_2$SO$_4$ in the
atmosphere is produced from oxidation reactions of SO$_2$ it would be expected that increased
concentrations of the latter would be associated with increased values for all the variables
associated with the NPF process. Contrary to this though, the relation of SO$_2$ concentrations with
NPF probability was found to be negative at all the sites in this study with available data. This
relation was relatively strong (R$^2$ > 0.50) in most areas with an increased significance at roadsides
compared to their respective rural sites. As this is a negative relation, this may indicate that SO$_2$ is
in sufficient concentrations for H$_2$SO$_4$ formation, thus not suppressing the occurrence of NPF
events, as well as showing that in increased concentrations, it is a more important factor (or
surrogate for a factor) in preventing the occurrence of NPF events within the urban environment, as
probably higher SO$_2$ is associated with increased co-emitted particle pollution and hence CS. The
growth rate on the other hand, presented mixed results and the significance of the relationships is
low in most cases, which makes these results untrustworthy. Finally, the relation of SO$_2$
concentrations with the formation rate was found to be positive at all sites but SPARU and FINRU
(which had the lowest concentrations across the sites of this study). The significance of this
relationship was rather low for all but the roadsides. This suggests that higher H$_2$SO$_4$ concentrations





favour increased formation rates (i.e. more particles can be formed), rather than necessarily
promoting nucleation itself because of the competing effect of condensation onto the pre-existing
particle population.

The normalised slopes $a_N^*$ were found to be more negative at the background sites compared to
their respective roadsides, as well as being less negative in the UK (where $SO_2$ is in greater
abundance) compared to the other sites with relatively significant relations. Plotting the average
$SO_2$ concentrations with the normalised slopes $a_N^*$ for the all sites (though not all had significant
relations), a positive relation with relatively high $R^2$ (when the extreme values from Marylebone
Road-UKRO are removed) is found which might indicate that while increased concentrations are a
negative factor in NPF event occurrence at a given site, in general the sites with higher $SO_2$
concentrations on average present higher probability for NPF events (Figures 7a and 7b). This
appears to be in agreement with Dall'Osto et al. (2018) who discussed the variable role of $SO_2$
depending on its concentrations. No significant relations were found for the values of $a_J^*$ as in most
cases these relations were rather weak.

**3.2.2    Nitrogen oxides or nitrogen dioxide ($NO_x$ or $NO_2$)**
$NO_x$ and $NO_2$ are directly associated with pollution, which can be a limiting factor for NPF events
as it increases the CS and may suppress the events (An et al., 2015), though with the reduction of
$SO_2$ concentrations achieved the last couple of decades, there is  possibility for oxidation products





of $NO_x$ to become an important component for NPF (Wang et al., 2020). For almost all sites (apart
from GRERU) with available data a negative relation between the NPF probability and $NO_x$ (or
$NO_2$) concentrations (depending on what data was available) was found. Similarly, for all the sites
but SPARU and GRERU, the correlations were strong with $R^2 > 0.43$. The rural background sites
had a weaker relation between the two variables compared to the urban sites, which is probably
associated with them having rather low concentrations of $NO_x$ (or $NO_2$) and variability, making the
variations of this factor less important. Growth rate had weaker correlations with $NO_x$ and different
trends between the sites, either being positive or negative. The variable effect of $NO_X$ on particle
growth, shifting HOMs' volatility, was previously discussed by Yan et al. (2020). While variability
was found for the background sites, all roadsides regardless of the strength of the relation had
positive relation between $NO_x$ and the growth rate. This may indicate the different components
associated with the growth process at each type of site which, as found in other studies can be
related to compounds associated with combustion processes that take place within the urban
environment (Guo et al., 2020; Wang et al., 2017a). The formation rate presents few cases of strong
relations, with variable trends (positive and negative). While much effort was made to isolate the
effect of NPF events by taking a shorter time frame before the event, the effect of local pollution is
still included, especially at the urban sites.

The normalised slopes do not provide a significant result for the relationship of this variable with
either the probability of the events or the formation rate. The only noteworthy points are the more





negative $a_N{}^*$ at the rural background sites compared to the roadsides in all the areas studied, which
shows the increased importance of a clean environment for NPF events to occur in areas where
condensable compounds are in lesser abundance, such as a rural environment. Additionally, the
negative slopes found at all the roadside sites, which increases the confidence that the events
extracted at the roadsides are not pollution incidents but NPF events.  However, it appears that
traffic pollution favours higher particle growth rates, although the components responsible for this
effect are unknown.

### 3.2.3    Ozone ($O_3$)
Ozone is typically the result of atmospheric photochemistry and is itself a source of hydroxyl
radical through photolysis, or ozonolysis of alkenes both during daytime and night-time (Fenske et
al., 2000) .  It might therefore be expected to act as an indicator of photochemical activity which
promotes the oxidation of $SO_2$ and VOCs.  Ozone concentrations may be directly related to the
solar radiation intensity as well as the pollution levels in the area studied, and $O_3$ is considered as a
positive factor in the occurrence of NPF events (Woo et al., 2001; Berndt et al., 2006). As for the
solar radiation, there is a strong relation between $O_3$ concentration and the probability for NPF
events. This positive relation was found to be stronger for the sites in northern Europe, while it was
not significant for the sites from southern Europe (Spanish sites and GRERU), possibly indicating
that $O_3$ is a less important factor at the southern sites. Specifically for the Spanish sites which have
the highest average concentrations of $O_3$ with some extreme values (Querol et al., 2017), the





relation of $O_3$ concentrations with the NPF probability presents a unique trend, having a clear peak
then a steady decline at both sites (though at different $O_3$ concentrations), which is also responsible
for the low correlations found (this trend seems to also occur at SPARU for the growth rate and to a
lesser extent for the formation rate as well, though for different $O_3$ concentration ranges). The
specific variability found at the Spanish sites was also studied by Carnerero et al., (2019). For sites
with a marked seasonal variation in ozone, associations with NPF may be artefactual due to
correlations with other variables such as temperature, RH and solar radiation.

Unlike the solar radiation though, the growth rate presents a negative relation at the sites where the
relation between these two variables was significant (UKRU, UKUB, DENUB and FINRU), which
might either be an indication of a polluted background that may have a negative effect in the growth
of the newly formed particles (though the trends found for $NO_x$ indicate differently) or specific
chemical processes which cannot be identified due to the lack of detailed chemical composition
data. A significant relation between $O_3$ and the formation rate was only found for a few sites
(though the trends become a lot clearer if some values are removed from the extreme lower or
higher end). This way the relations become strong, but positive, for some areas and negative for
some others without any clear trend (type or location of the site, $O_3$ concentrations etc.). No clear
relation between these two variables was found as the sites with strong relation have both positive
and negative relationships and as a result no confident conclusions can be drawn.





As the correlations found were strong the normalised slopes for NPF probability, when plotted
against the average concentrations of $O_3$, present a negative correlation with relatively high $R^2$
(0.64), indicating that the $O_3$ is a more important factor in the occurrence of NPF events when in
lower concentrations (Figure 8). Finally, though with a low level of confidence for the southern
sites, the $a_N^*$ were smaller at the southern sites compared to those in the north, up to one order of
magnitude between the FINRU (furthest north rural background) and GRERU (furthest south rural
background).

**3.2.4       Organic compounds**
**3.2.4.1   Particulate organic carbon (OC)**
Organic carbon (OC) compounds are considered as components with importance in the growth of
newly formed particles, with a role that becomes increasingly important as the size of the particles
becomes larger (Nieminen et al., 2010; Zhang et al., 2012; Shrivastava et al., 2017). Particulate OC,
the data for which are available in the present study, can be associated with pollution, especially in
the urban environment. Only a few of the sites of the present study were found to have a strong
negative relationship ($R^2 > 0.50$) of particulate OC with the NPF probability (UKUB, UKRO and
DENRU). Regardless though of the strength of this relation, all other sites (apart from FINRU) had
a negative relationship between these two variables as well, consistent with increased
concentrations of particulate OC being associated with increased pollution, which is a suppressing
factor in the occurrence of NPF events. Growth rate on the other hand was found to have a slight





positive relation ($R^2 > 0.40$) for most of the sites. This relation appeared to be stronger (higher $R^2$)
at the roadsides with available data compared to their respective rural background sites. The relation
between particulate OC and the growth rate was positive at all the sites with available data
regardless of their significance showing that, despite its effect in the occurrence of NPF events, it is
still a favourable variable for the growth of the particles. The formation rate was found to have a
significant relation with particulate OC concentrations at half of the sites with available data
(UKUB, UKRO, DENRU, DENRO).

The normalised slopes for this variable did not present any noteworthy relations with either the type
of site or the concentrations of OC at a given site.

**3.2.4.2    Volatile organic compounds (VOCs)**
Many volatile organic compounds have been found to be associated with the NPF process. Benzene,
toluene, ethylbenzene, m+p-xylene, o-xylene and trimethylbenzenes have been reported to be able
to form Highly Oxygenated Organic Molecules (HOMs) in flow tubes (Wang et al., 2017a; Molteni
et al., 2018), which may act as contributors to particle nucleation and/or growth. Xylenes, and to a
lesser extent trimethylbenzenes, are the most efficient at forming HOMs. Benzene and toluene are
less efficient and will form more volatile HOMs. These HOMs may all be too volatile to form new
particles, though this is not yet confirmed. Chamber studies involving $H_2SO_4$ and trimethylbenzene
oxidation products were associated with high formation rates when measuring $J_{1.5}$ (Metzger et al.,





2010). All these HOMs though will be sufficiently involatile to contribute to particle growth. Those
with higher oxygen content or carbon number will be classed as LVOC and if they dimerise, they
will form ELVOC (Bianchi et al., 2019). Monoterpenes can also form HOMs which drive both the
formation (Ehn et al., 2014; Riccobono et al., 2014) and growth (Tröstl et al., 2016), while isoprene
can act as a sink for hydroxyl radical (Kiendler-Scharr et al., 2009) and is not as effective in HOM
and secondary organic aerosol formation compared to monoterpenes (McFiggans et al., 2019).

Volatile organic compound data were available for three of the sites of this study (Table S2). Two
of the sites with VOC data were from the rural background and the roadside in the UK. Most of the
compounds are associated with combustion sources and were found to have a negative relationship
with NPF event occurrence at both sites, with high $R^2$ in most cases. Additionally, isoprene, which
may have either biogenic or anthropogenic sources (Wagner and Kuttler, 2014) was also found to
have a negative relationship with NPF event occurrence at Marylebone Road-UKRO, though with
low $R^2$. This result is in line with the VOCs being strongly correlated with particulate OC (which
presented a negative relation with NPF event probability, as discussed in Section 3.2.4.1), as well as
with the CS (which also presented a negative relationship with NPF event probability, as mentioned
in Section 3.2.6), further associating these compounds with combustion emissions.

Growth rate was found to have a positive relationship with VOCs in almost all cases for both UK
sites. Few exceptions were found (with only 1,3 butadiene having a relatively high $R^2$) which





presented a negative relationship with the growth rate in rural Harwell-UKRU. Finally, the
formation rate presented a different behaviour between the two sites. At Harwell-UKRU, the
relationship was unclear in most cases, with a group of VOCs presenting a negative relationship
with the formation rate (ethane, ethene, propane, 1,3 butadiene, toluene, ethylbenzene, o-xylene and
1,2,4 trimethylbenzene – with $R^2 > 0.40$), two VOCs presented a rather clear positive relationship
with the formation rate (iso-pentane and 2-methylbenzene) and the rest of the VOCs had an unclear
relationship. At Marylebone Road-UKRO though, VOCs presented a positive relationship with the
formation rate (for particles of diameter 16 nm). This is probably due to the fact that these VOCs
are associated with pollution emissions (as mentioned earlier) and though a smaller time window
was chosen to avoid including the effect of the morning rush hour traffic, this is very difficult in the
traffic polluted environment of Marylebone Road-UKRO.

As Hyytiälä (FINRU) is a rural background site far from the direct effect of combustion emissions,
different VOCs were measured, which mainly originate from biogenic sources rather than
anthropogenic ones. The results were mixed and less clear compared to those from the UK sites
(mainly due to the smaller dataset), and three groups were found depending on their relationship
with NPF probability. The first group, including acetonitrile, acetic acid and Methyl Ethyl Ketone
(MEK) presented a slight positive relation. The second group presented a negative relation, with the
VOCs in this group being MEK, monoterpenes, benzene, isoprene and toluene (only the last two
have $R^2 > 0.50$). Finally, the third group included VOCs that presented a peak and then a decline for





higher concentrations including methanol, and acetone. Two groups of VOCs were found
depending on their relationship with the growth rate. The ones with a positive relation being
methanol, acetonitrile, acetone, acetic acid, isoprene, MEK, monoterpenes and toluene, while
acetaldehyde, MEK and benzene had a negative relationship, with relatively high $R^2$ in most cases.
Finally, the results with the formation rate were unclear with only a handful presenting weak
positive (methanol, acetic acid and benzene) or negative (MEK) relations that do not appear to be
significant. The normalised slopes cannot be used for VOCs as there are very few sites with
available data.

**3.2.5    Sulphate ($SO_4^{-2}$)**
Sulphate ($SO_4^{2-}$) is a major secondary constituent of aerosols. Secondary $SO_4^{2-}$ aerosols largely arise
from either gas phase reaction between $SO_2$ and OH, or in the aqueous phase by the reaction of $SO_2$
and $O_3$ or $H_2O_2$, or $NO_2$ (Hidy et al., 1994). In environments where $SO_4^{2-}$ chemistry is dominant
(i.e. remote areas), $SO_4^{2-}$ and ammonium (bi) sulphate (($NH_4)_2SO_4$ and $NH_4HSO_4$) particles are a
large relative contributor to aerosol mass, while this contribution is lower in environments where
other emissions are also significant (i.e. urban areas where the secondary $NO_3^-$ relative contribution
is a lot higher). While not well established, a possible relation of $SO_4^{2-}$-containing compounds and
variables of NPF events was found in previous studies (Beddows et al., 2015; Minguillón et al.,
2015; Wang et al., 2017b). In the present study, only a few sites had $SO_4^{2-}$ data available, for $PM_1$
(FINRU), $PM_{2.5}$ (Danish sites) or $PM_{10}$ (rest of the sites). While this data cannot be considered as





directly associated with the ultrafine particles, for two sites with available AMS data for ultrafine
particles, the direct comparison between $SO_4^{2-}$ aerosol in PM and in the range of particles of about
50 nm, very high correlations were found (results not included). For all the sites with available data
the NPF probability presented a negative relation. The significance of this relations was found to be
relatively high ($R^2 > 0.50$) only for background sites (apart from GERRU, which has rather low
concentrations and probably different mechanisms for the NPF events). Similarly, the growth rate
presented a more significant relation ($R^2 > 0.40$) for the same background sites (apart from FINRU),
though this relationship was found to be positive at all sites regardless of its significance. Finally,
the formation rate did not present a clear trend as it was found to have both negative and positive
relations for different sites. This relation was significant only for two rural sites (UKRU and
DENRU) and as a result no assumptions can be made.

The normalised slopes cannot be used for any analysis on sulphate as the measurements available
are from different particle sizes.

**3.2.6    Gaseous ammonia (NH₃)**
Ammonia (NH₃) can be an important compound in the nucleation process according to the ternary
theory (Napari et al., 2002). It was found that elevations in NH₃ concentrations can lead to
elevations to NPF rate (Lehtipalo et al., 2018) and it was also found to be an important factor for
NPF event occurrence even when stronger bases are present in high concentrations (Glasoe et al.,





2015). No significant variation was found though between event and non-event days in a previous
study in Harwell-UKRU (Bousiotis et al., 2019). Data for gaseous ammonia were only available for
Harwell-UKRU and presented a positive relation with NPF probability, until reaching a peak point.
Further increase in $NH_3$ concentrations presented a decline with NPF probability, which might be
due to its association with increased pollution levels. Interesting though is that it presented a clear
positive relation with both the growth rate (though it also appears to decline at high concentrations)
and the formation rate.

**3.2.7    Condensation sink (CS)**
The CS is a measure of the rate at which molecules will condense onto pre-existing aerosols
(Lehtinen et al., 2003). It is highly dependent on the number and size of the particles in the
atmosphere and as a result it is expected to be affected by both the local emissions within the urban
environment as well as the formation and growth of the particles due to NPF events. As a result, for
the specific metric a time frame before the events are in full development was chosen (05:00 to
10:00 LT) to avoid including the effect of the NPF events and provide a picture of the atmospheric
conditions that preceded the NPF events. With this data, the NPF probability presented very strong
relations with the condensation sink. Two groups of sites were found though; those which had a
positive relation and those with a negative relation. In the first group are the sites in Germany and
Greece while all others had a negative relation. This grouping follows the trend between the
countries, the sites of which presented a greater (the ones with the positive slopes) or smaller CS on





NPF event days, though it is unknown what causes this behaviour (at the German sites and GREUB
it may be associated with the very high formation rates on NPF event days). While the slopes from
this analysis cannot be used for direct comparisons, a trend was found for which the slopes were
more positive or negative at the rural sites compared to their respective roadsides, which might
indicate the greater importance of the variability of the CS at the rural sites in the occurrence of
NPF events.

The growth rate was positively correlated with the CS for most of the sites, with strong relations
(high $R^2$) for about half of them. As the CS is a metric of pre-existing particles, it is also associated
with the level of pollution in a given area. The increased significance and slope found at the rural
sites probably indicates the importance of enhanced presence of condensable compounds in a
cleaner environment, which in many cases are associated with the moderate presence of pollution.
The formation rate was also found to have a positive relation with the CS. This relation was more
significant at the roadsides of this study, a result which to some extent is biased by the presence of
increased traffic emissions found in the timeframe chosen. While to an extent, increased presence of
condensable compounds can be favourable for greater formation rates, this result should be
considered with great caution.

The normalised slopes $a_N^*$ followed a similar trend as those found with the initial analysis. These
slopes were found to be more positive or negative, depending on the trend of the given area, at the





rural sites compared to their roadsides. The urban background sites did not always have a uniform
behaviour (though in UK, Denmark and Finland these were between the rural site and the roadside),
due to their more diverse character compared to the other two types of sites.

**3.3      Association of the Effect of the Variables**
The Pearson correlation coefficients for the variables studied on each site are found in Table S1.
The relatively strong relation between the solar radiation, temperature and $O_3$ found, as well as their
anticorrelation with the RH may lead to the conclusion that not all these factors play a role in NPF
events, but their visible effect is the result of their relationship with each other. There is a similar
case with the association of the CS and $NO_x$ (or $NO_2$), and OC, as well as $SO_2$, especially at urban
sites. However, the factors affect different outcomes differently, as for example the solar radiation
intensity does not seem to be as important a factor for the growth rate as temperature, or $O_3$ does not
seem to be strongly associated with either the formation or the growth rate. This is further
established by the fact that some of these variables do not correlate well at the southern sites, but
still appear to be associated with either the probability of NPF events or the growth or nucleation
rate. The effects of all of these factors have been demonstrated in both laboratory and atmospheric
studies in the past and were discussed earlier in this paper. By the analysis provided in the present
study, the effect of each of these variables is further established, providing an association of each
one of these variables with either the formation or the growth mechanism. However, RH does not
seem to be a consistent factor in any mechanism, and it appears that its effect is dependent on





location specific conditions, although it was the variable with the most consistent relation with NPF
event probability at almost all sites.

**3.4     Relationship to a previous  multi-station European study**
The findings of our study in respect of the background sites show many similarities with the
conclusions drawn in the previous multi-station study in Europe by Dall'Osto et al. (2018) despite
the two studies using several different sampling stations as well as some in common. Both studies
point towards the influence of variables such as solar radiation and CS upon the occurrence of NPF
events. The previous study suggested that different compounds participate in the growth of the
particles, depending on the area considered. Thus, for northern and southern sites the growth of the
particles is suggested to be driven mainly by organic compounds, while for the sites in central
Europe sulphate plays a more important role. These findings are confirmed by the present study, as
the growth rate was found to correlate better with organic compounds for the rural sites in Finland
and Greece, while $SO_4^{2-}$ presented a stronger relation with the growth rate for the Danish and
German sites (the latter presented high slope values but low $R^2$ due to a decline at higher $SO_4^{2-}$
concentrations, probably associated with NPF events being suppressed by increased pollution). The
growth of the particles at the rural background site in the UK, characterised as "Overlap" in the
previous study, was found to be strongly associated with both organic compounds and sulphate,
consistent with it being in the central group.





The seasonality of NPF events at northern sites was hard to explain in the previous study, and the
possible effect of low temperature was considered. In the present study, the Finnish background
sites presented a double-peak relation of NPF probability with temperature, with one of the peaks
being below zero degrees. This might point to the possibility of different compounds driving the
events for different temperature ranges, as well as the increased nucleation rate of $H_2SO_4$ at lower
temperatures (Kirkby et al., 2011; Yan et al., 2018), which makes the occurrence of NPF events
more probable at lower temperatures in a region with low $SO_2$ concentrations.

**4.     CONCLUSIONS**
More than 85 site-years of data from 16 sites from six countries in Europe were analysed for NPF
events. A total of 1950 NPF events with consequent growth of the newly formed particles were
extracted and with the use of binned linear regression, the relation between three variables
associated with NPF events (NPF event probability, formation and growth rate) with meteorological
conditions and atmospheric composition was studied. Among the meteorological conditions, solar
radiation, temperature and atmospheric pressure presented a positive relation with NPF event
occurrence, and either promoting the formation or growth rate. Relative humidity presented a
negative relation with NPF event probability which in most cases was associated with it being a
limiting factor on particle formation at higher values. Wind speed on the other hand presented
variable results, appearing to depend on the location of the sites rather than their type. This shows
that while wind speed can be a factor in NPF event occurrence, the origin of the incoming air



masses also plays a very important role. In most cases, meteorological conditions appeared to be
more important factors in NPF event occurrence at rural sites compared to urban sites, suggesting
that NPF events are driven more by them at this type of site. Additionally, while some
meteorological variables appeared to play a crucial role in the occurrence of NPF events, this role
appears to become less important at higher values when a positive relation was found (or lower
when a negative relation was found).

The results for the levels of atmospheric pollutants presented a more interesting picture as most of
these, which appear to be either directly or indirectly associated with the NPF process were found to
have negative relations with NPF probability. This is probably due to the fact that increased
concentrations of such compounds are associated with more polluted conditions, which are a
limiting factor in the occurrence of NPF events, as was found with the negative relation between the
CS and NPF probability in most cases. Thus, $SO_2$, $NO_x$ (or $NO_2$), particulate OC and $SO_4^{2-}$
concentrations were negatively correlated with NPF probability in most cases. Average $SO_2$
concentrations though appeared to correlate positively with the normalised NPF event probability
slopes with relatively significant correlation, indicating that while increasing concentrations have a
negative impact in the occurrence of NPF events at a given site, in general sites with higher $SO_2$
concentrations have higher probability for NPF events. On the other hand though, these compounds
in many cases had a positive relation (not always though with high significance) with the other
variables considered. Thus, particulate OC (and VOCs where data were available) and $SO_4^{2-}$





consistently had a positive relation with the growth rate, while $SO_2$ was positively associated with
both the formation and growth rate in most cases. Finally, $O_3$ was positively correlated with NPF
event probability at all sites in this study, though it presented variable results with the other two
variables. As with some meteorological conditions it was found that at sites with increased
concentrations of $O_3$, its importance as a factor was decreased, which to an extent can be related
with high CS associated with peak summer $O_3$ days in southern Europe.

The present study attempts to explain the effect of several meteorological and atmospheric variables
on the occurrence and development of NPF events, by using a large-scale dataset.  It should be
noted that the variables considered are in many cases inter-related (e.g. temperature and RH) and
this complicates considerably the interpretation in terms of causal factors. Large datasets are very
useful in providing with more uniform results by removing the possible bias of short period
extremities, which may lead to wrong assumptions. Following from this, the importance of a high-
resolution measurement network, both site and timewise is underlined, as it can help in elucidating
the mechanisms of new particle formation in the real atmosphere.

**DATA ACCESSIBILITY**
Data supporting this publication are openly available from the UBIRA eData repository at
https://doi.org/https://doi.org/10.25500/edata.bham.00000491



**AUTHOR CONTRIBUTIONS**
The study was conceived and planned by RMH who also contributed to the final manuscript, and
DB who also carried out the analysis and prepared the first draft of the manuscript. AM, JKN, CN,
JVN, HP, NP, AA, GK, SV and KE have provided with the data for the analysis. JB provided help
with analysis of the data. FDP provided advice on the analysis. MDO, XQ and TP contributed to the
final manuscript.

**COMPETING INTERESTS**
The authors have no conflict of interests.

**ACKNOWLEDGMENTS**
This work was supported by the National Centre for Atmospheric Science funded by the U.K.
Natural Environment Research Council (R8/H12/83/011).



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





**TABLE LEGENDS**

**Table 1:**     Location and data availability of the sites.

**Table 2:**     Frequency and formation rate of NPF events for the sites of the study.

**Table 3:**     Normalised slopes (non-normalised for growth rate), $R^2$ and p-values (- for values >0.05) for the relation between meteorological conditions and NPF event variables.

**Table 4:**     Normalised slopes (non-normalised for growth rate), $R^2$ and p-values (- for values >0.05) for the relation between atmospheric composition variables and NPF event variables.

**FIGURE LEGENDS**

**Figure 1:**     Map of the sites of the present study.

**Figure 2:**     Relation of average downward incoming solar radiation (K↓) and normalised slopes $a_N^*$ for the sites of the present study.

**Figure 3:**     Normalised slopes $a_J^*$ for K↓ for the sites of the present study (*UK sites are calculated with solar irradiance).

**Figure 4a:**     Relation of average relative humidity and normalised slopes $a_N^*$ for the sites of the present study.

**Figure 4b:**     Relation of average relative humidity and normalised slopes $a_N^*$ for the sites of the present study (SPAUB not included).

**Figure 5:**     Relation of average temperature and normalised slopes $a_N^*$ for the sites of the present study.

**Figure 6:**     Normalised slopes $a_J^*$ for temperature for the sites of the present study.

**Figure 7a:**     Relation of average $SO_2$ concentrations and normalised slopes $a_N^*$ for the sites of the present study.

**Figure 7b:**     Relation of average $SO_2$ concentrations and normalised slopes $a_N^*$ for the sites of the present study (UKRO not included).





**Figure 8:**  Relation of average $O_3$ concentrations and normalised slopes $a_N^*$ for the sites of the
present study.





**Table 1:** Location and data availability of the sites.

| Site | Location | Available data | Meteorological data location | Data availability | Reference |
|---|---|---|---|---|---|
| UKRU | Harwell Science Centre, Oxford, 80 km W of London, UK (51o 34' 15" N; 1o 19' 31" W) | SMPS (16.6 - 604 nm, 76.5% availability), NOx, SO2, O3, OC, SO42-, gaseous ammonia | On site | 2009 - 2015 | Charron et al., 2013 |
| UKUB | North Kensington, 4 km W of London city centre, UK (51o 31' 15" N; 0o 12' 48" W) | SMPS (16.6 - 604 nm, 83.3% availability), NOx, SO2, O3, OC, SO42- | Heathrow airport | 2009 - 2015 | Bigi and Harrison, 2010 |
| UKRO | Marylebone Road, London, UK (51o 31' 21" N; 0o 9' 16" W) | SMPS (16.6 - 604 nm, 74.3% availability), NOx, SO2, O3, OC, SO42- | Heathrow airport | 2009 - 2015 | Charron and Harrison, 2003 |
| DENRU | Lille Valby, 25 km W of Copenhagen, (55o 41' 41" N; 12o 7' 7" E) (2008 – 6/2010) Risø, 7 km north of Lille Valby, (55° 38' 40" N; 12° 5' 19" E) (7/2010 – 2017) | DMPS and CPC (5.8 - 700 nm, 68.3% availability), NOx, SO2, O3, OC, SO42- | H.C. Ørsted – Institute station | 2008 – 2017 | Ketzel et al., 2004 |
| DENUB | H.C. Ørsted – Institute, 2 km NE of the city centre, Copenhagen, Denmark (55o 42' 1" N; 12o 33' 41" E) | DMPS and CPC (5.8 - 700 nm, 61.4% availability), NOx, O3 | On site | 2008 – 2017 | Wang et al., 2010 |
| DENRO | H.C. Andersens Boulevard, Copenhagen, Denmark (55o 40' 28" N; 12o 34' 16" E) | DMPS and CPC (5.8 - 700 nm, 65.7% availability), NOx, SO2, O3, OC, SO42- | H.C. Ørsted – Institute station | 2008 – 2017 | Wang et al., 2010 |
| GERRU | Melpitz, 40 km NE of Leipzig, Germany (51o 31' 31.85" N; 12o 26' 40.30" E) | TDMPS with CPC (4.8 - 800 nm, 87.2% availability), OC, SO42- | On site | 2008 – 2011 | Engler et al., 2007 |
| GERUB | Tropos, 3 km NE from the city centre of Leipzig, Germany (51o 21' 9.1" N; 12o 26' 5.1" E) | TDMPS with CPC (3 - 800 nm, 90.4% availability) | On site | 2008 – 2011 | Costabile et al., 2009 |
| GERRO | Eisenbahnstraße, Leipzig, Germany (51o 20' 43.80" N; 12o 24' 28.35" E) | TDMPS with CPC (4 - 800 nm, 68.3% availability) | Tropos station | 2008 – 2011 | Birmili et al., 2016 |
| FINRU | Hyytiälä, 250 km N of Helsinki, Finland (61o 50' 50.70" N; 24o 17' 41.20" E) | TDMPS with CPC (3 – 1000 nm, 98.2% availability), NOx, SO2, O3, VOCs | On site | 2008 – 2011 & 2015 – 2018 | Aalto et al., 2001 |
| FINUB | Kumpula Campus 4 km N of the city centre, Helsinki, Finland (60o 12' 10.52" N; 24o 57' 40.20" E) | TDMPS with CPC (3.4 - 1000 nm, 99.7% availability) | On site | 2008 – 2011 & 2015 – 2018 | Järvi et al., 2009 |
| FINRO | Mäkelänkatu street, Helsinki, Finland (60o 11' 47.57" N; 24o 57' 6.01" E) | DMPS (6 - 800 nm, 90.0% availability), NOx, O3 | Pasila station and on site | 2015 – 2018 | Hietikko et al., 2018 |
| SPARU | Montseny, 50 km NNE from Barcelona, Spain (41o 46' 45" N; 2o 21' 29" E) | SMPS (9 – 856 nm, 53.7% availability), NO2, SO2, O3 | On site | 2012 - 2015 | Dall'Osto et al., 2013 |
| SPAUB | Palau Reial, Barcelona, Spain (41o 23' 14" N; 2o 6' 56" E) | SMPS (11 – 359 nm, 88.1% availability), NO2, SO2, O3 | On site | 2012 – 2015 | Dall'Osto et al., 2012 |
| GRERU | Finokalia, 70 km E of Heraklion, Greece (35o 20' 16.8" N; 25o 40' 8.4" E) | SMPS (8.77 - 849 nm, 85.0% availability), NO2, O3, OC | On site | 2012 – 2018 | Kalkavouras et al., 2017 |





| | | | | | |
|---|---|---|---|---|---|
| **GREUB** | "Demokritos", 12 km NE from the city centre, Athens, Greece (37o 59' 41.96" N; 23o 48' 57.56" E) | SMPS (10 – 550 nm, 88.0% availability) | On site | 2015 – 2018 | Mølgaard et al., 2013 |



**Table 2:** Frequency and formation rate of NPF events for the sites of the study.

| Site | Frequency of NPF events (%) | $J_{10}$ (N cm$^{-3}$ s$^{-1}$) |
|---|---|---|
| UKRU | 7.0 | 8.69E-03* |
| UKUB | 7.0 | 1.42E-02* |
| UKRO | 6.1 | 3.75E-02* |
| DENRU | 7.9 | 2.57E-02 |
| DENUB | 5.8 | 2.40E-02 |
| DENRO | 5.4 | 8.07E-02 |
| GERRU | 17.1 | 9.18E-02 |
| GERUB | 17.5 | 1.02E-01 |
| GERRO | 9.0 | 1.38E-01 |
| FINRU | 8.7 | 1.19E-02 |
| FINUB | 5.0 | 2.49E-02 |
| FINRO | 5.1 | 6.94E-02 |
| SPARU | 12 | 1.54E-02 |
| SPAUB | 13.1 | 2.12E-02 |
| GRERU | 6.5 | 4.90E-03 |
| GREUB | 8.5 | 4.41E-02 |

\* $J_{16}$ calculated





**Table 3:** Normalised slopes (non-normalised for growth rate), $R^2$ and p-values (- for values >0.05) for the relation between meteorological conditions and NPF event variables.

| Downward shortwave solar radiation K↓ (W m$^{-2}$) | | | | | | | | | |
|---|---|---|---|---|---|---|---|---|---|
| **Site** | $a_N^*$ (W$^{-1}$ m$^2$) | $R^2$ | p | $a_G$ | $R^2$ | p | $a_J^*$ (W$^{-1}$ m$^2$) | $R^2$ | p | Average |
| **UKRU\*** | 1.21E-03 | 0.94 | <0.001 | 6.53E-05 | 0.11 | - | 6.28E-04 | 0.93 | <0.001 | 443 |
| **UKUB\*** | 6.81E-04 | 0.90 | <0.001 | -8.26E-05 | 0.10 | - | 1.49E-04 | 0.19 | - | 448 |
| **UKRO\*** | 8.69E-04 | 0.98 | <0.001 | -7.75E-06 | 0.00 | - | 2.66E-04 | 0.64 | <0.005 | 464 |
| **DENRU** | 2.22E-03 | 0.88 | <0.001 | 4.24E-04 | 0.20 | - | 1.38E-03 | 0.64 | <0.001 | 115 |
| **DENUB** | 1.87E-03 | 0.91 | <0.001 | 1.47E-04 | 0.03 | - | 8.98E-04 | 0.48 | <0.01 | 115 |
| **DENRO** | 2.46E-03 | 0.95 | <0.001 | 1.27E-04 | 0.01 | - | 6.77E-04 | 0.50 | <0.005 | 117 |
| **GERRU** | 2.87E-03 | 0.98 | <0.001 | 9.88E-04 | 0.72 | <0.01 | 1.45E-03 | 0.81 | <0.001 | 130 |
| **GERUB** | 3.18E-03 | 0.97 | <0.001 | 7.28E-04 | 0.51 | <0.005 | 1.53E-03 | 0.69 | <0.001 | 114 |
| **GERRO** | 2.40E-03 | 0.95 | <0.001 | -5.89E-04 | 0.09 | - | 9.95E-04 | 0.59 | <0.005 | 114 |
| **FINRU** | 2.63E-03 | 0.76 | <0.001 | 1.01E-03 | 0.57 | <0.01 | 2.04E-03 | 0.82 | <0.001 | 91.5 |
| **FINUB** | 1.38E-03 | 0.37 | - | 1.81E-04 | 0.08 | - | 8.99E-04 | 0.25 | - | 111 |
| **FINRO** | 1.76E-03 | 0.59 | <0.005 | 9.15E-04 | 0.34 | <0.005 | 4.45E-04 | 0.03 | - | 114 |
| **SPARU** | 3.46E-04 | 0.35 | <0.05 | 5.68E-04 | 0.13 | - | 1.97E-03 | 0.74 | <0.001 | 162 |
| **SPAUB** | 5.92E-04 | 0.58 | <0.05 | 6.98E-04 | 0.23 | - | 1.58E-03 | 0.81 | <0.001 | 180 |
| **GRERU** | 4.10E-04 | 0.52 | <0.001 | 7.14E-04 | 0.55 | <0.001 | -6.30E-04 | 0.05 | - | 201 |
| **GREUB** | 3.49E-04 | 0.31 | - | -1.10E-04 | 0.02 | - | 8.97E-04 | 0.34 | <0.05 | 183 |

\* Global solar irradiation measurements in kJ m$^{-2}$

| Relative Humidity (%) | | | | | | | | | |
|---|---|---|---|---|---|---|---|---|---|
| **Site** | $a_N^*$ (%$^{-1}$) | $R^2$ | p | $a_G$ | $R^2$ | p | $a_J^*$ (%$^{-1}$) | $R^2$ | p | Average |
| **UKRU** | -5.89E-02 | 0.85 | <0.001 | 1.69E-03 | 0.02 | - | -3.35E-02 | 0.85 | <0.001 | 79.7 |
| **UKUB** | -3.42E-02 | 0.94 | <0.001 | 8.23E-03 | 0.24 | - | -5.66E-03 | 0.19 | - | 75.3 |
| **UKRO** | -5.09E-02 | 0.85 | <0.001 | 7.03E-03 | 0.25 | - | -1.49E-02 | 0.46 | <0.05 | 74.5 |
| **DENRU** | -3.90E-02 | 0.95 | <0.001 | 9.42E-03 | 0.74 | <0.001 | 5.45E-04 | 0.00 | - | 75.7 |
| **DENUB** | -3.14E-02 | 0.94 | <0.001 | 3.64E-03 | 0.06 | - | 2.57E-03 | 0.00 | - | 75.7 |
| **DENRO** | -3.64E-02 | 0.95 | <0.001 | -1.21E-02 | 0.22 | - | -3.91E-03 | 0.10 | - | 75.7 |
| **GERRU** | -5.08E-02 | 0.88 | <0.001 | -1.30E-02 | 0.72 | <0.001 | -2.46E-02 | 0.91 | <0.001 | 81.9 |
| **GERUB** | -5.35E-02 | 0.86 | <0.001 | -6.34E-03 | 0.67 | <0.001 | -2.25E-02 | 0.86 | <0.001 | 78.7 |
| **GERRO** | -2.83E-02 | 0.90 | <0.001 | 3.98E-03 | 0.05 | - | -1.72E-02 | 0.81 | <0.001 | 78.7 |
| **FINRU** | -4.48E-02 | 0.94 | <0.001 | -7.07E-03 | 0.65 | <0.001 | -2.16E-02 | 0.87 | <0.001 | 80.1 |
| **FINUB** | -5.89E-02 | 0.95 | <0.001 | 1.04E-02 | 0.26 | - | -6.52E-03 | 0.18 | - | 76.5 |
| **FINRO** | -3.34E-02 | 0.92 | <0.001 | -1.47E-03 | 0.01 | - | 7.39E-03 | 0.10 | - | 71.1 |
| **SPARU** | -1.54E-02 | 0.90 | <0.001 | -4.67E-03 | 0.08 | - | -7.12E-03 | 0.14 | - | 66.4 |
| **SPAUB** | -4.84E-02 | 0.93 | <0.001 | 2.43E+02 | 0.50 | <0.01 | -9.83E-03 | 0.19 | - | 69.2 |
| **GRERU** | -7.72E-03 | 0.22 | - | 1.06E-02 | 0.06 | - | -1.83E-01 | 0.15 | - | 70.0 |
| **GREUB** | -1.42E-02 | 0.62 | <0.001 | 2.83E-03 | 0.06 | - | 4.85E-04 | 0.00 | - | 60.5 |





| Temperature (°C) | | | | | | | | | |
|---|---|---|---|---|---|---|---|---|---|
| Site | $a_N^*$ (°C$^{-1}$) | $R^2$ | p | $a_G$ | $R^2$ | p | $a_J^*$ (°C$^{-1}$) | $R^2$ | p | Average |
| UKRU | 1.10E-01 | 0.93 | <0.001 | 7.85E-02 | 0.94 | <0.001 | 8.72E-02 | 0.84 | <0.001 | 10.6 |
| UKUB | 9.04E-02 | 0.98 | <0.001 | 1.39E-01 | 0.96 | <0.001 | 6.34E-02 | 0.73 | <0.005 | 11.8 |
| UKRO | 8.22E-02 | 0.98 | <0.001 | 3.51E-02 | 0.52 | <0.05 | 4.32E-02 | 0.44 | <0.05 | 12.1 |
| DENRU | 6.68E-02 | 0.83 | <0.001 | 1.54E-02 | 0.08 | - | 6.68E-02 | 0.92 | <0.001 | 9.80 |
| DENUB | 2.50E-02 | 0.45 | <0.05 | 2.40E-02 | 0.33 | - | 3.05E-02 | 0.45 | <0.05 | 9.82 |
| DENRO | 6.64E-02 | 0.88 | <0.001 | 3.51E-03 | 0.00 | - | 2.96E-02 | 0.58 | <0.005 | 10.0 |
| GERRU | 7.27E-02 | 0.92 | <0.001 | 5.65E-02 | 0.92 | <0.001 | 5.37E-02 | 0.93 | <0.001 | 10.3 |
| GERUB | 8.20E-02 | 0.93 | <0.001 | 3.38E-02 | 0.62 | <0.001 | 4.28E-02 | 0.54 | <0.005 | 11.1 |
| GERRO | 5.08E-02 | 0.89 | <0.001 | -3.33E-03 | 0.00 | - | 1.61E-02 | 0.11 | - | 11.1 |
| FINRU | -2.01E-02 | 0.17 | - | 1.13E-01 | 0.79 | <0.001 | 4.27E-02 | 0.72 | <0.001 | 4.79 |
| FINUB | -4.21E-03 | 0.00 | - | 7.42E-02 | 0.83 | <0.001 | 1.67E-02 | 0.28 | - | 6.52 |
| FINRO | 6.24E-02 | 0.65 | <0.005 | 9.28E-02 | 0.87 | <0.001 | -1.09E-02 | 0.05 | - | 7.72 |
| SPARU | -2.51E-02 | 0.41 | <0.05 | 1.23E-01 | 0.92 | <0.001 | 9.11E-02 | 0.71 | <0.001 | 13.9 |
| SPAUB | -3.43E-03 | 0.02 | - | 6.67E-02 | 0.66 | <0.005 | 1.18E-02 | 0.08 | - | 18.2 |
| GRERU | -4.66E-02 | 0.75 | <0.001 | 1.74E-01 | 0.75 | <0.001 | -9.45E-02 | 0.47 | <0.05 | 18.2 |
| GREUB | -1.00E-02 | 0.25 | - | 4.67E-02 | 0.62 | <0.005 | -2.85E-02 | 0.20 | - | 17.6 |

| Wind Speed (m s$^{-1}$) | | | | | | | | | |
|---|---|---|---|---|---|---|---|---|---|
| Site | $a_N^*$ (m$^{-1}$ s) | $R^2$ | p | $a_G$ | $R^2$ | p | $a_J^*$ (m$^{-1}$ s) | $R^2$ | p | Average |
| UKRU | 5.72E-02 | 0.20 | - | -3.04E-02 | 0.07 | - | 6.87E-03 | 0.00 | - | 3.96 |
| UKUB | 1.72E-01 | 0.87 | <0.001 | -1.91E-01 | 0.71 | <0.001 | 3.56E-03 | 0.00 | - | 4.16 |
| UKRO | 6.34E-02 | 0.19 | - | 3.21E-02 | 0.02 | - | 7.28E-02 | 0.45 | <0.005 | 4.14 |
| DENRU | 1.08E-01 | 0.88 | <0.001 | -2.33E-01 | 0.74 | <0.001 | 1.28E-01 | 0.44 | <0.01 | 4.17 |
| DENUB | 1.50E-01 | 0.90 | <0.001 | -3.33E-02 | 0.10 | - | 8.31E-02 | 0.19 | - | 4.17 |
| DENRO | 1.65E-01 | 0.89 | <0.001 | -1.51E-01 | 0.49 | <0.001 | 9.08E-03 | 0.00 | - | 4.16 |
| GERRU | -1.06E-01 | 0.57 | <0.005 | -2.26E-01 | 0.83 | <0.001 | -5.32E-03 | 0.00 | - | 2.58 |
| GERUB | -1.27E-01 | 0.52 | <0.01 | -1.41E-01 | 0.60 | <0.005 | -3.32E-02 | 0.04 | - | 2.33 |
| GERRO | -2.40E-01 | 0.56 | - | -2.54E-01 | 0.38 | - | -1.30E-01 | 0.22 | - | 2.33 |
| FINRU | 1.62E-01 | 0.63 | <0.005 | -1.29E-01 | 0.16 | <0.05 | 7.99E-02 | 0.07 | - | 1.31 |
| FINUB | -3.17E-02 | 0.08 | - | 7.26E-02 | 0.20 | <0.05 | -9.74E-02 | 0.17 | - | 3.43 |
| FINRO | 8.62E-02 | 0.51 | <0.05 | -1.60E-01 | 0.32 | <0.05 | -1.86E-01 | 0.32 | - | 4.26 |
| SPARU | -2.20E-02 | 0.02 | - | 3.80E-01 | 0.31 | - | 5.74E-02 | 0.02 | - | 0.94 |
| SPAUB | 2.90E-01 | 0.93 | <0.001 | 7.71E-02 | 0.24 | - | -5.90E-02 | 0.05 | - | 2.05 |
| GRERU | 4.37E-02 | 0.54 | <0.001 | 1.01E-01 | 0.36 | <0.005 | 1.73E-03 | 0.00 | - | 6.06 |
| GREUB | -1.13E-01 | 0.47 | <0.01 | -1.88E-01 | 0.50 | <0.005 | -3.78E-02 | 0.01 | - | 1.87 |





| Atmospheric Pressure (mbar) | | | | | | | | | |
|---|---|---|---|---|---|---|---|---|---|
| Site | $a_N$* (mbar$^{-1}$) | $R^2$ | p | $a_G$ | $R^2$ | p | $a_J$* (mbar$^{-1}$) | $R^2$ | p | Average |
| UKRU | 4.26E-02 | 0.83 | <0.005 | 3.93E-02 | 0.58 | <0.005 | 2.95E-02 | 0.47 | <0.05 | 1007.7 |
| UKUB | 1.90E-02 | 0.50 | - | 1.17E-02 | 0.05 | <0.05 | 4.16E-03 | 0.04 | - | 1011.7 |
| UKRO | 6.33E-02 | 0.95 | <0.001 | -1.21E-01 | 0.40 | - | -2.98E-02 | 0.17 | - | 1012 |
| GERRU | 5.10E-02 | 0.97 | - | 8.95E-02 | 0.85 | <0.001 | 2.16E-02 | 0.21 | - | 1007.0 |
| GERUB | 6.27E-02 | 0.97 | - | 4.00E-02 | 0.76 | - | 2.00E-02 | 0.37 | <0.05 | 995.5 |
| GERRO | 4.57E-02 | 0.79 | - | -9.61E-02 | 0.43 | - | -2.80E-02 | 0.21 | - | 995.5 |
| FINRU | 3.46E-02 | 0.88 | <0.001 | 2.90E-02 | 0.57 | <0.001 | 1.05E-02 | 0.14 | - | 985.1 |
| FINUB | 2.61E-02 | 0.55 | <0.005 | -3.57E-03 | 0.02 | - | 4.38E-03 | 0.05 | - | 1004.4 |
| FINRO | 4.91E-02 | 0.70 | - | -2.67E-02 | 0.17 | - | 1.43E-02 | 0.26 | - | 1008.8 |
| SPARU | -2.02E-02 | 0.09 | - | 4.79E-02 | 0.14 | - | 2.89E-02 | 0.08 | - | 939.3 |
| SPAUB | -2.83E-02 | 0.44 | <0.05 | 1.86E-02 | 0.08 | - | 1.68E-02 | 0.21 | - | 1006.3 |
| GRERU | 6.00E-02 | 0.46 | <0.001 | -1.50E-01 | 0.73 | - | 8.14E-02 | 0.33 | - | 1014.5 |
| GREUB | 9.42E-03 | 0.10 | <0.05 | -1.00E-01 | 0.71 | - | 1.58E-02 | 0.04 | - | 1015.7 |



**Table 4:** Normalised slopes (non-normalised for growth rate), $R^2$ and p-values (- for values >0.05) for the relation between atmospheric composition variables and NPF event variables.

| | | | | $SO_2$ ($\mu g \ m^{-3}$) | | | | | | |
|---|---|---|---|---|---|---|---|---|---|---|
| Site | $a_N*$ ($\mu g^{-1} \ m^3$) | $R^2$ | p | $a_G$ | $R^2$ | p | $a_J*$ ($\mu g^{-1} \ m^3$) | $R^2$ | p | Average |
| UKRU | -1.97E-01 | 0.38 | <0.05 | -6.17E-02 | 0.02 | - | 3.30E-01 | 0.06 | - | 1.64 |
| UKUB | -2.57E-01 | 0.62 | <0.001 | 1.93E-02 | 0.00 | - | 4.18E-01 | 0.40 | - | 2.04 |
| UKRO | -1.03E-01 | 0.82 | <0.001 | 6.90E-02 | 0.34 | <0.01 | 8.43E-02 | 0.77 | <0.001 | 7.46 |
| DENRU | -9.77E-01 | 0.53 | <0.05 | 2.84E+00 | 0.37 | - | 4.38E-01 | 0.09 | - | 0.52 |
| DENRO | -4.20E-01 | 0.91 | <0.001 | 6.42E-01 | 0.54 | <0.005 | 5.66E-01 | 0.62 | <0.001 | 0.97 |
| FINRU | -5.66E-01 | 0.05 | - | -1.42E+00 | 0.19 | - | -6.30E-02 | 0.00 | - | 0.09 |
| SPARU | -3.62E-01 | 0.74 | <0.001 | -1.33E-01 | 0.02 | - | -3.55E-02 | 0.01 | - | 0.95 |
| SPAUB | -2.93E-02 | 0.04 | - | 4.12E-01 | 0.59 | - | 1.07E-01 | 0.29 | - | 1.99 |

| | | | | $NO_x$ or $NO_2$ (ppb) | | | | | | |
|---|---|---|---|---|---|---|---|---|---|---|
| Site | $a_N*$ ($ppb^{-1}$) | $R^2$ | p | $a_G$ | $R^2$ | p | $a_J*$ ($ppb^{-1}$) | $R^2$ | p | Average |
| UKRU | -4.99E-02 | 0.67 | <0.005 | 4.52E-02 | 0.58 | <0.05 | -4.51E-02 | 0.70 | <0.005 | 11.7 |
| UKUB | -8.75E-03 | 0.83 | <0.001 | -3.97E-04 | 0.00 | - | -1.09E-02 | 0.43 | <0.05 | 53.6 |
| UKRO | -3.22E-03 | 0.72 | <0.001 | 1.44E-03 | 0.39 | <0.05 | 2.19E-03 | 0.66 | <0.001 | 299 |
| DENRU | -9.41E-02 | 0.43 | <0.005 | -4.89E-03 | 0.00 | <0.001 | -6.47E-02 | 0.55 | <0.01 | 5.42 |
| DENUB | -4.99E-02 | 0.68 | <0.001 | 2.85E-02 | 0.26 | - | 8.55E-04 | 0.00 | - | 10.5 |
| DENRO | -5.10E-03 | 0.75 | <0.001 | 1.10E-02 | 0.69 | <0.001 | 8.33E-03 | 0.88 | <0.001 | 68.5 |
| FINRU | -7.27E-01 | 0.54 | <0.001 | -2.74E-01 | 0.11 | - | 1.95E-01 | 0.05 | - | 0.72 |
| FINRO | -6.24E-03 | 0.68 | <0.001 | 1.70E-03 | 0.12 | - | 3.25E-03 | 0.03 | - | 88.1 |
| SPARU* | -1.53E-02 | 0.05 | - | 2.54E-02 | 0.01 | - | 1.25E-01 | 0.21 | - | 3.26 |
| SPAUB* | -2.59E-02 | 0.62 | <0.005 | 2.23E-02 | 0.70 | <0.001 | 2.57E-03 | 0.01 | - | 31.4 |
| GRERU* | 3.01E-01 | 0.19 | - | -1.40E+00 | 0.75 | <0.001 | 5.23E-01 | 0.13 | - | 0.52 |

* $NO_2$ measurements



| $O_3$ (ppb) | | | | | | | | | |
|---|---|---|---|---|---|---|---|---|---|
| Site | $a_N$* (ppb$^{-1}$) | $R^2$ | p | $a_G$ | $R^2$ | p | $a_J$* (ppb$^{-1}$) | $R^2$ | p | Average |
| UKRU | 2.27E-02 | 0.88 | <0.001 | -4.89E-02 | 0.53 | <0.005 | -3.53E-03 | 0.01 | - | 54.4 |
| UKUB | 1.37E-02 | 0.87 | <0.001 | -3.45E-02 | 0.68 | <0.001 | -5.95E-03 | 0.05 | - | 39.3 |
| UKRO | 7.46E-02 | 0.95 | <0.001 | -1.06E-02 | 0.09 | - | -2.44E-02 | 0.63 | <0.005 | 16.2 |
| DENRU | 4.97E-02 | 0.92 | <0.001 | -1.32E-02 | 0.15 | - | 1.23E-02 | 0.08 | - | 30.1 |
| DENUB | 5.85E-02 | 0.84 | <0.001 | -1.69E-02 | 0.58 | - | 2.77E-02 | 0.32 | <0.05 | 28.2 |
| DENRO | 6.42E-02 | 0.51 | <0.05 | 1.39E-02 | 0.03 | - | 3.24E-02 | 0.91 | <0.05 | 31.1 |
| FINRU | 6.76E-02 | 0.77 | <0.05 | -4.23E-02 | 0.60 | - | 3.92E-02 | 0.37 | <0.05 | 27.4 |
| FINRO | 2.38E-02 | 0.91 | <0.001 | 6.11E-03 | 0.24 | - | -1.83E-02 | 0.29 | - | 37.1 |
| SPARU | 1.57E-02 | 0.02 | - | 4.34E-02 | 0.11 | - | 1.31E-02 | 0.31 | - | 75.9 |
| SPAUB | 7.99E-03 | 0.38 | <0.05 | -5.83E-03 | 0.30 | - | -1.13E-03 | 0.01 | - | 54.9 |
| GRERU | 7.55E-03 | 0.04 | - | 3.68E-02 | 0.17 | - | -3.01E-02 | 0.15 | - | 49.5 |

| Particulate Organic Carbon (µg m$^{-3}$) | | | | | | | | | |
|---|---|---|---|---|---|---|---|---|---|
| Site | $a_N$* (µg$^{-1}$ m$^3$) | $R^2$ | p | $a_G$ | $R^2$ | p | $a_J$* (µg$^{-1}$ m$^3$) | $R^2$ | p | Average |
| UKRU | -3.30E-02 | 0.00 | - | 1.13E+00 | 0.42 | <0.005 | 2.13E-01 | 0.16 | - | 1.96 |
| UKUB | -2.76E-01 | 0.59 | <0.005 | 6.63E-01 | 0.58 | <0.05 | 2.19E-01 | 0.55 | <0.05 | 3.63 |
| UKRO | -3.78E-01 | 0.89 | <0.001 | 8.12E-01 | 0.57 | <0.005 | 4.60E-01 | 0.75 | <0.001 | 6.24 |
| DENRU | -4.44E-01 | 0.75 | <0.001 | 2.24E-01 | 0.11 | - | -3.17E-01 | 0.68 | <0.01 | 1.48 |
| DENRO | -7.80E-02 | 0.11 | - | 1.10E+00 | 0.77 | <0.005 | 4.02E-01 | 0.81 | <0.005 | 2.59 |
| GERRU | -1.26E-01 | 0.24 | - | 1.35E-01 | 0.09 | - | 3.14E-02 | 0.03 | - | 2.18 |
| FINRU | 2.27E-02 | 0.00 | - | 3.39E-01 | 0.60 | <0.005 | -3.46E-01 | 0.16 | - | 1.78 |
| GRERU | -2.08E-01 | 0.11 | - | 7.87E-01 | 0.41 | <0.05 | 8.94E-01 | 0.11 | - | 1.58 |

| Sulphate (µg m$^{-3}$) | | | | | | | | | |
|---|---|---|---|---|---|---|---|---|---|
| Site | $a_N$* (µg$^{-1}$ m$^3$) | $R^2$ | p | $a_G$ | $R^2$ | p | $a_J$* (µg$^{-1}$ m$^3$) | $R^2$ | p | Average |
| UKRU[1] | -2.62E-01 | 0.57 | <0.001 | 7.34E-01 | 0.77 | <0.001 | 7.99E-01 | 0.44 | <0.05 | 1.97 |
| UKUB[1] | -3.57E-01 | 0.89 | <0.001 | 9.28E-01 | 0.44 | <0.01 | 9.72E-01 | 0.16 | - | 1.58 |
| UKRO[1] | -6.05E-02 | 0.24 | - | 3.04E-01 | 0.34 | <0.05 | -6.22E-02 | 0.04 | - | 1.98 |
| DENRU[2] | -7.81E-01 | 0.34 | <0.05 | 1.02E+00 | 0.60 | <0.05 | -1.03E+00 | 0.63 | <0.01 | 0.52 |
| DENRO[2] | -8.23E-01 | 0.28 | - | 1.99E+00 | 0.22 | - | 2.82E-01 | 0.12 | - | 0.55 |
| GERRU[1] | -3.37E-02 | 0.00 | - | 5.89E-01 | 0.11 | - | -4.89E-02 | 0.01 | - | 0.92 |
| FINRU[3] | -1.18E+00 | 0.65 | <0.001 | 2.35E-01 | 0.09 | - | -2.53E-01 | 0.17 | - | 1.02 |





[1] Measurements in $PM_{10}$

[2] Measurements in $PM_{2.5}$

[3] Measurements in $PM_1$

| Condensation Sink ($s^{-1}$) | | | | | | | | | |
|---|---|---|---|---|---|---|---|---|---|
| **Site** | $a_N$* (s) | $R^2$ | p | $a_G$ | $R^2$ | p | $a_J$* (s) | $R^2$ | p | **Average** |
| **UKRU** | -2.28E+02 | 0.72 | <0.001 | 2.64E+02 | 0.60 | <0.001 | 7.58E+01 | 0.22 | - | 3.38E-03 |
| **UKUB** | -1.66E+02 | 0.78 | <0.001 | 2.49E+02 | 0.41 | <0.05 | 1.73E+02 | 0.35 | <0.05 | 7.41E-03 |
| **UKRO** | -4.03E+01 | 0.75 | <0.001 | 2.33E+01 | 0.18 | - | 8.94E+01 | 0.91 | <0.001 | 2.12E-02 |
| **DENRU** | -4.48E+01 | 0.91 | <0.001 | 6.90E+01 | 0.49 | <0.05 | 5.37E+01 | 0.24 | - | 9.46E-03 |
| **DENUB** | -3.78E+01 | 0.75 | <0.001 | 3.58E+01 | 0.25 | - | 1.55E+01 | 0.56 | <0.005 | 1.42E-02 |
| **DENRO** | -1.06E+01 | 0.73 | <0.001 | 2.53E+01 | 0.56 | <0.005 | 2.72E+01 | 0.79 | <0.001 | 3.10E-02 |
| **GERRU** | 1.54E+02 | 0.86 | <0.001 | 1.33E+02 | 0.56 | <0.001 | 6.67E+01 | 0.63 | <0.001 | 7.02E-03 |
| **GERUB** | 3.59E+01 | 0.56 | <0.005 | 3.63E+01 | 0.17 | - | 4.74E+01 | 0.75 | <0.001 | 9.11E-03 |
| **GERRO** | 3.89E+01 | 0.22 | <0.05 | -2.21E+01 | 0.03 | <0.005 | 3.54E+01 | 0.45 | <0.005 | 1.20E-02 |
| **FINRU** | -1.80E+02 | 0.59 | <0.005 | 4.01E+02 | 0.74 | <0.001 | 4.98E+01 | 0.10 | - | 2.32E-03 |
| **FINUB** | -1.51E+02 | 0.63 | <0.005 | 8.14E+01 | 0.31 | - | 2.01E+02 | 0.41 | <0.05 | 6.34E-03 |
| **FINRO** | -6.99E+01 | 0.77 | <0.001 | -1.56E+01 | 0.05 | - | 2.42E+02 | 0.83 | <0.001 | 8.96E-03 |
| **SPARU** | -2.15E+02 | 0.65 | <0.005 | 1.86E+01 | 0.00 | - | 8.60E+01 | 0.47 | <0.05 | 5.49E-03 |
| **SPAUB** | -1.18E+02 | 0.65 | <0.005 | 3.74E+01 | 0.38 | <0.05 | 9.51E+01 | 0.52 | <0.01 | 1.00E-02 |
| **GRERU** | 4.33E+00 | 0.00 | - | 2.86E+02 | 0.70 | <0.001 | 1.77E+02 | 0.56 | <0.005 | 4.66E-03 |
| **GREUB** | 1.64E+02 | 0.65 | <0.001 | 9.31E+01 | 0.28 | <0.05 | 1.73E+02 | 0.83 | <0.001 | 7.55E-03 |







**Figure 1:** Map of the sites of the present study.





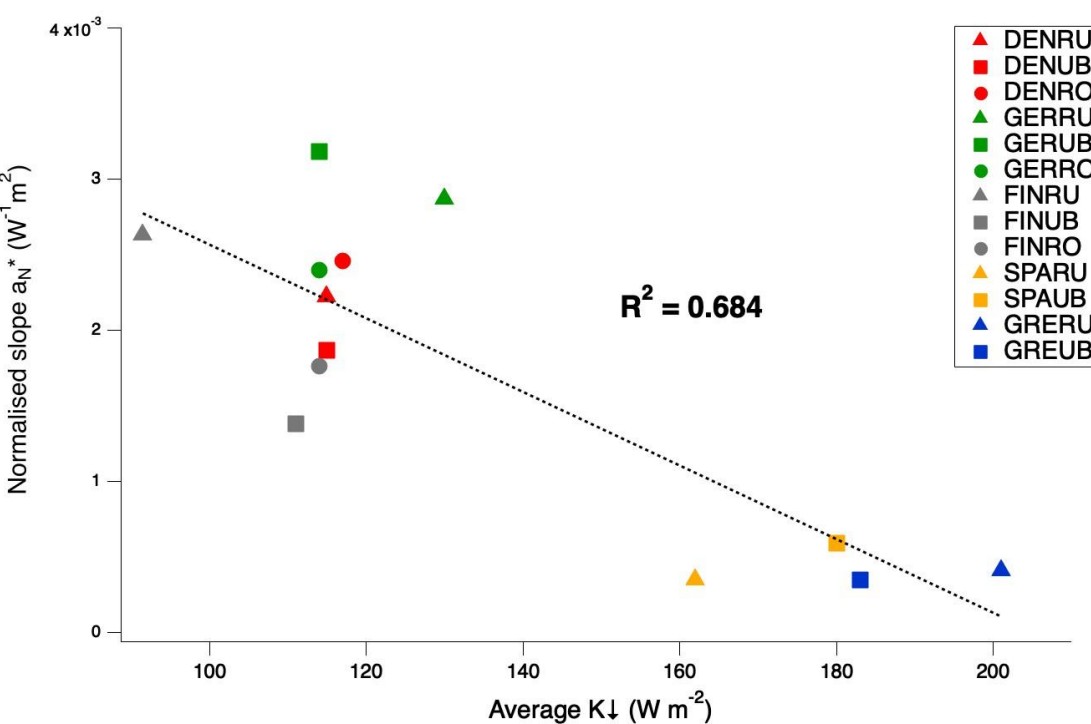

**Figure 2:** Relation of average downward incoming solar radiation (K↓) and normalised slopes $a_N$* for the sites of the present study.

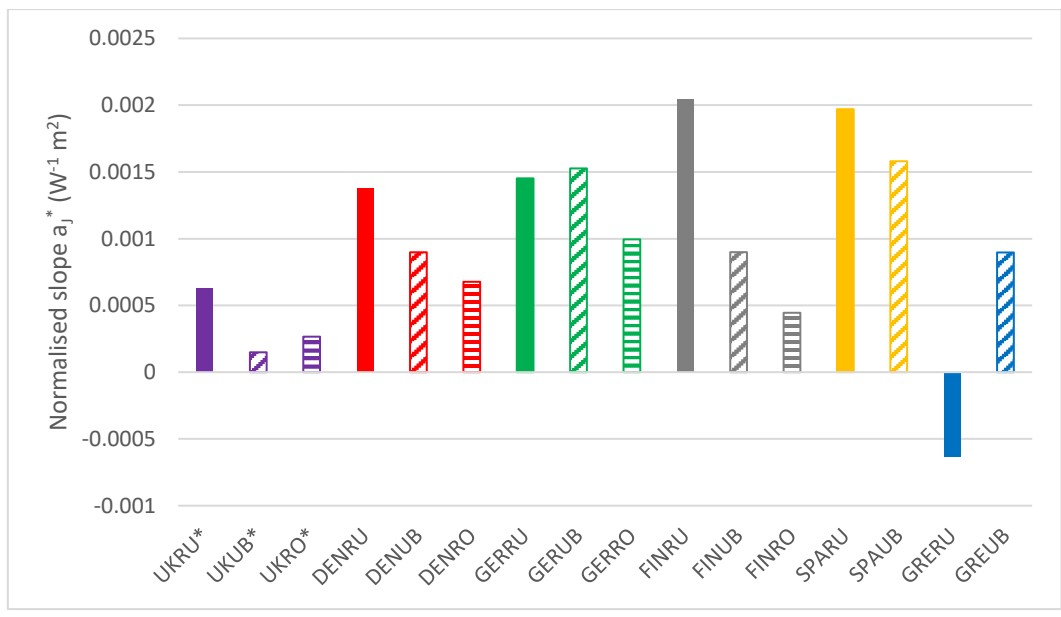

**Figure 3:** Normalised slopes $a_J$* for K↓ for the sites of the present study (*UK sites are calculated with solar irradiance).





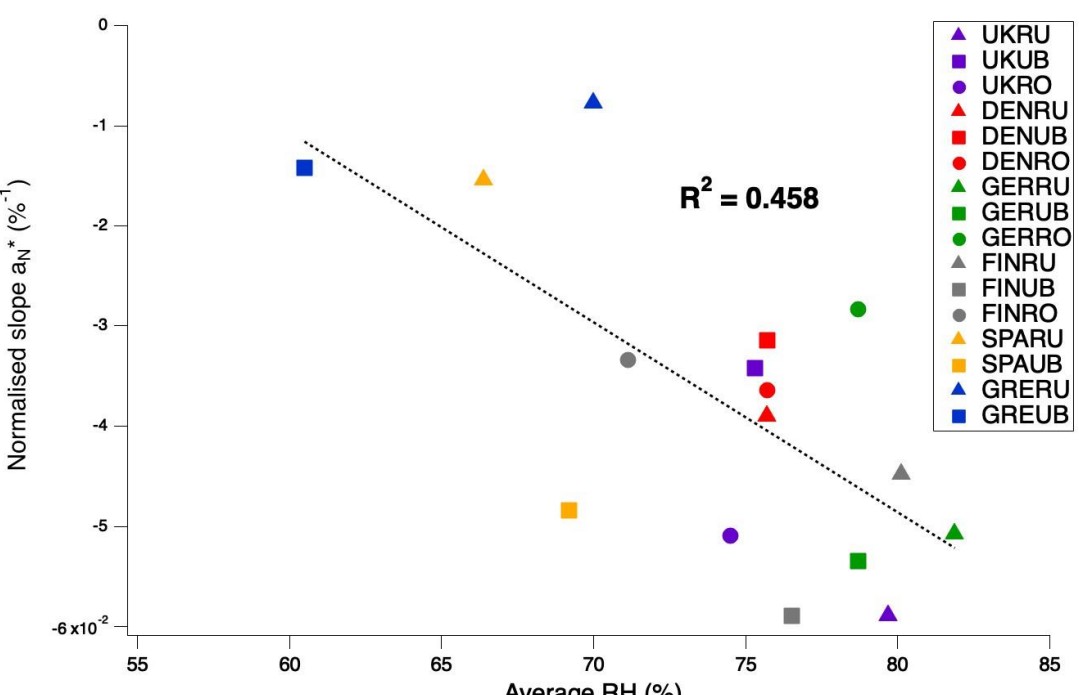

**Figure 4:** Relation of average relative humidity and normalised slopes $a_N^*$ for the sites of the present study.

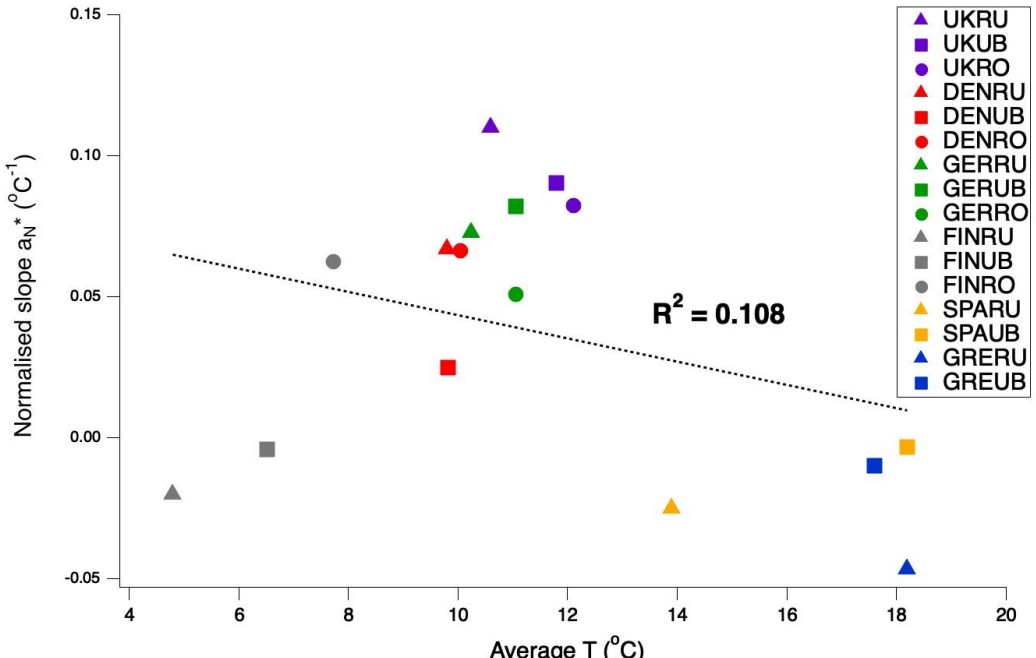

**Figure 5:** Relation of average temperature and normalised slopes $a_N^*$ for the sites of the present study.





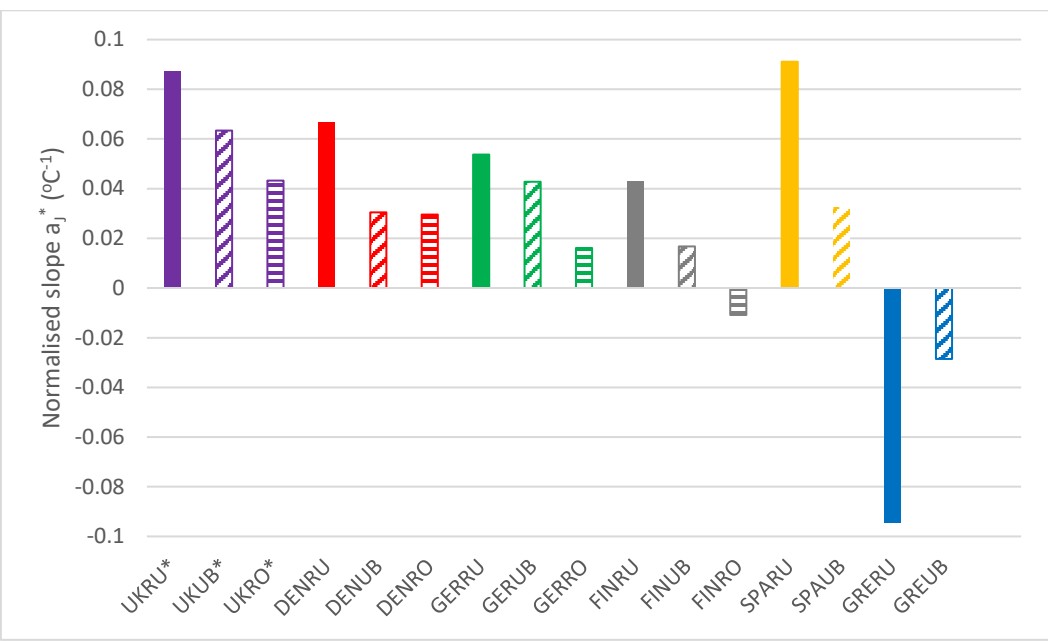

**Figure 6:** Normalised slopes $a_J^*$ for temperature for the sites of the present study.





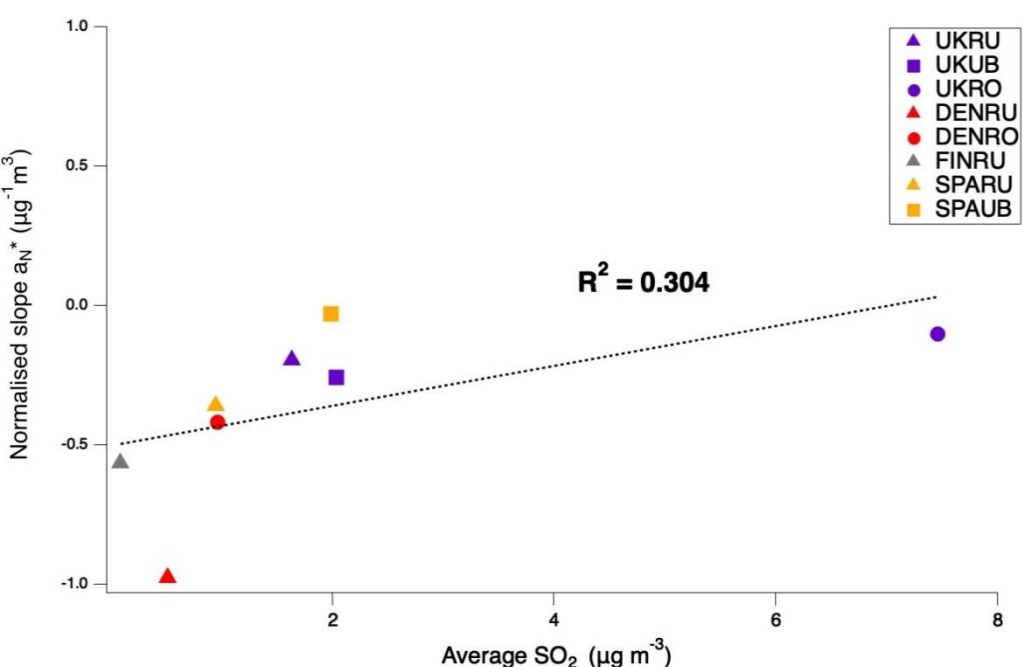

**Figure 7a:** Relation of average $SO_2$ concentrations and normalised slopes $a_N^*$ for the sites of the present study.

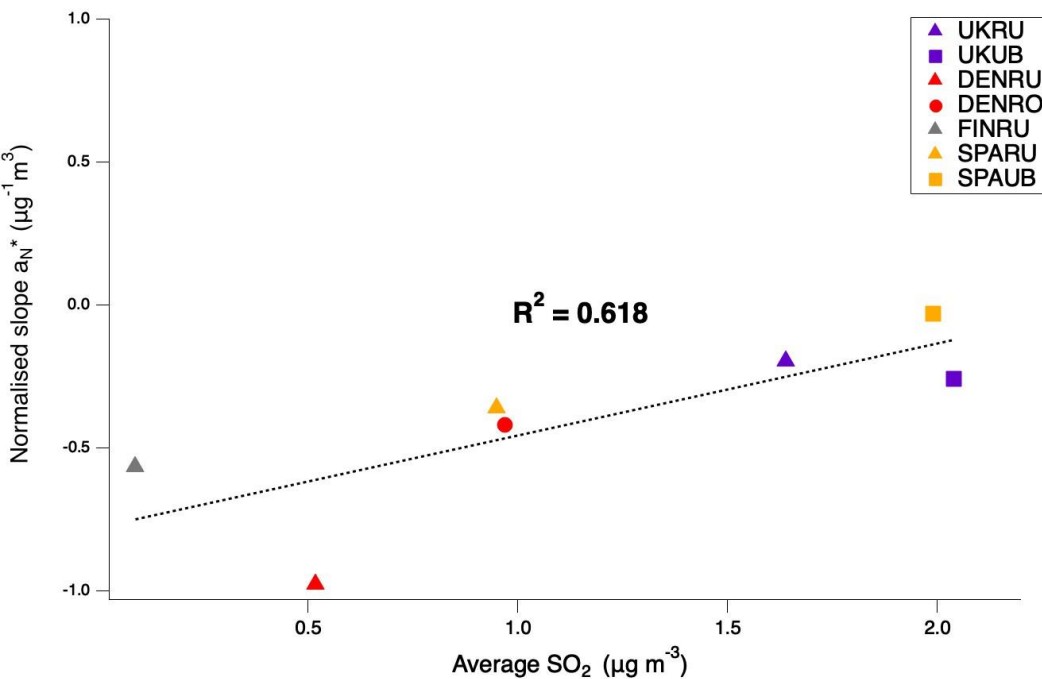

**Figure 7b:** Relation of average $SO_2$ concentrations and normalised slopes $a_N^*$ for the sites of the present study (UKRO not included).





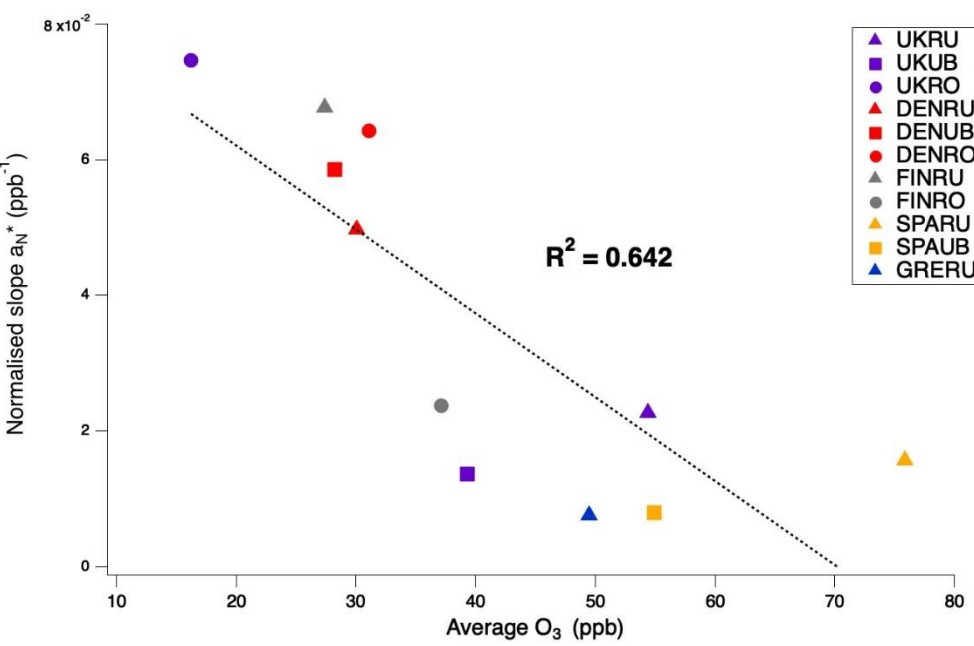

**Figure 8:** Relation of average $O_3$ concentrations and normalised slopes $a_N{}^*$ for the sites of the present study.
