# Peer review of "The Effect of Meteorological Conditions and Atmospheric Composition in the Occurrence and Development of New Particle Formation (NPF) Events in Europe"

_Atmospheric Chemistry and Physics, 2020_

## Referee Comment (RC2) · Anonymous Referee #2 · 26 Sep 2020

General comments

The focus of this study is to investigate the effect of meteorological conditions and atmospheric composition on the occurrence of new particle formation (NPF) events at 16 sites (rural, urban background and roadside) located in 6 European countries. The results are based on more than 85 years of meteorological and atmospheric composition data. The authors are using a binned linear regression to find correlations between parameters such as windspeed, temperature, pressure, or solar radiation intensity, ozone or volatile organic compounds mixing ratios to name a few, and the occurrence of NPF,

particle growth and the formation rates. This is an interesting study and of interest to the community, however the following comments should be addressed before publishing.

On many occasions the authors claim that certain variables are weakly or strongly correlated but do not provide any numbers or figure to support these statements. Please provide references to the exact figures in the manuscript or in the supplemental material (SM). This is an issue reappearing throughout the manuscript. Following on that, Figure S1 in SM contains many figures and only one caption. These figures are not marked with a number/letter. Please consider adding individual numbering (or introduce letters) so the respective figures corresponding to individual sites when being discussed in the manuscript could be easily found in SM.

It looks like the authors use terms frequency of NPF occurrence and NPF probability interchangeably. NPF probability doesn't really fit here since you do not predict NPF events. However, the NPF probability term is explained in the text and in the equation (line 191; also please number equations). In results, however, the authors are using term frequency of NPF occurrence (line 245). Please clarify, review the explanation in text and use the correct term throughout the manuscript. I assume what you want to use is NPF frequency.

I understand you identify the number of days with NPF according to the method by Dal Maso et al. (2005) with additional certain criteria. It would be good to report the numbers of NPF events for each site (and season?). Please explain what days with "relevant data" are when calculating the frequency.

If I understood correctly to calculate the frequency, you divide the number of days identified as NPF event-days by all days that you have data available or "relevant(?)" data available? I am curious how does the frequency changes when when you use the number of all days with all data and not only with "relevant data available"? It would be good to mention this number somewhere in the manuscript or in the SM? Following on

the above, please explain what is in e.g. line 191 "available data" and "given group"?

More detail on the site selection criteria would be helpful. Do these sites belong to a network? How were these sites selected? In Line 120: the authors mention "geographical region and type of environment". I suggest adding more description on the sites (e.g. in SM), their characteristic and typical meteorological conditions, e.g. features they have in common/differences, number of NPF studied and identified.

Having this extensive dataset, I encourage the authors to discuss variability e.g. seasonal, site to site/regional. A figure where you plot frequency of NPF occurrence or number of NPF events for each station in each season (e.g. bar plot?) on (y axis), for each site type (x axis) would be helpful. Exploring e.g. seasonal variability would add value to the paper.

Where there any limitations of the study? If yes, these should be discussed. Further, errors should be included.

Are there general trends for these three site types? Maybe you could discuss these more. It would be helpful to highlight (e.g. text in bold) data in Table S1 e.g. significant correlations.

What is the importance of the result of this study? The authors could discuss it more e.g. in conclusion. I feel that is missing in the current version.

Please improve the language. It is critical to make the text more concise and clearer. It is hard to follow the line of thoughts at points. There are some repetitions and long sentences that could be shortened (e.g. lines: 76-82, 107-111, 325-330).

Specific comments

Line 38-62: in the abstract the authors could also mention: 85 years of data; how good these correlations are (r2)' mention "meteorological conditions" e.g. such as . . .

Line 42: "except at very clean air sites" – more information is needed to this statement.

[Figure]

Something is missing. Please review or explain.

Line 54: What "higher values" means there exactly? Provide a number.

Line 61-62: you could give these values in brackets

Line 97: "negative effect" on?

Line 99: "average conditions"? What does it mean here?

Line 107-111: hard to follow, please review and shorten

Line 121: please add references to the studies you refer to

Line 122: NPF probability? Or NPF occurrence? As mentioned above, probability doesn't really fit here

Line 124: I suggest calling this section: "2. Methods", 2.1 as is. 2.2 as is or similar. This way you can remove 2.2 Methods so it does not appear twice. In 2.1 the authors could mention which cities/countries/sites were used; which meteorological and atmospheric composition variables did you use in this study already at this point. Which stations had a full set of data and which only some etc. Maybe also mention which are dependent and which independent variables. And what do you consider relevant data days, what do you mean by available data: e.g. in line 189. Please be more specific upfront. You could also add information on how these sites were chosen? Any criteria you applied to select these? Are they belong to a network? Are they similar or different in any respect?

Line 127: I feel that the number of events (1950) is already a result so it should go to the result section and not methods. Also, it is mentioned before the description of the NPF selection method itself.

Line 131: it is also referring to the result. I suggest moving this sentence to the result section.

[Figure]

Line 136-143: please add more details to the approach taken in this study. What "Ia" exactly refers to and which additional criteria was used (line 142).

Line 137: add size range of the nucleation mode you consider in your study

Line 139: you could mention confidence level in the brackets

Line 151: add "respectively" after "particles". You could already mention there Fuchs correction factor and keep it explained below.

Line 149: Formulas need to be numbered

Line 188: given group? please explain

Line 191: Again: I am not sure I follow this equation: what are these groups? Is it just a number of days with NPF that was accompany by all relevant data? From explanation you seem only to take days with NPF that were accompanied by relevant data.

Line 196: low significance? Please give a number

Line 212: extreme values? Please give a number

Line 239: Results and Discussion? You could include here sentence in line 126: You mention 1950 events studied, could you provide information on how many were identified? It would be helpful if authors mention that in the paper a summary of data can be found in the manuscript and in the SM data/results for individual sites is presented.

Line 245: what is relevant data? Please explain more clearly in methods section and refer to it. Diagnostic features – wouldn't these be better in methods?

Line 252: "slopes and $R2$" please use correct terms for these or more careful description

Line 261: very strong? Please provide references to the exact figures in the figures/supplemental material when discussing results

Line 279: low? Please give a number and refer to the figure. Also, you placed all

figures in the SM under Figure 1. Maybe it would be better for the reader to have them split into different figure numbers or a,b,c,d? This way it would be easier to find the one you describe at the very moment.

Line 296: reference?

Line 301-303: why? Could you explain? When describing results maybe worth mentioning these for various site specifics? Anything in common?

Line 369: which factors remain constant?

Line 377: reference?

Line: 398: maximum? Low?

Line 420: Ethesian: add few words what these are could be added

Table 3: what is a "p value"? has it been defined somewhere? In tables: the authors could use bold text to highlight significant correlations? So these patterns/trends could be clearly seen?

Figures: no need to mention in each caption "of the present studies"

Line 433: you could already mention here which pollutants (such as...) are studied and described in the upcoming sections.

Line 752: "at higher values"?

Line 755: "meteorological conditions" such as?

Line 756-757: is that the only explanation? How about chemistry/composition at such type of site? Anything else that might play a role?

Line 782-783: seems out of place here; it would be more suitable at the beginning of conclusion section or removed.
* * *
[Figure]

2020.

---

## Author Response (AR1)

**MS No.:** acp-2020-555
**Title:** The Effect of Meteorological Conditions and Atmospheric Composition in the Occurrence
and Development of New Particle Formation (NPF) Events in Europe
**Author(s):** Dimitrios Bousiotis et al.
                **RESPONSE TO REVIEWERS**
The authors thank the reviewers for their insightful comments and have made many modifications
in response, and to enhance the clarity of the paper.
**Anonymous Referee #1**
A couple of decades ago, a number of studies tried to link meteorological variables and gas phase
pollutants with NPF. In some cases, the analysis was concise enough to produce evidence that a
certain physical parameter played a role in NPF at a specific site.
Since then, studies - mostly chamber-based - have provided evidence on the ruling mechanisms of
nucleation and subsequent growth of newly formed particles. These are mainly related to the
concentration of low vapor pressure compounds such as sulfuric acid, or ELVOC as well as agents
that could stabilize the former (ammonia, amines, iodine) suggesting that NPF is dictated mainly by
gas phase chemistry rather than meteorology. However, other parameters such as the ones
investigated in this study, play a secondary yet important role. Therefore a summary of observations
from European, or even better global sites, is always welcome. During the past 15 years more than
20 compilations of results related to atmospheric NPF have been published, the majority of which
are summarized by Kerminen et al., 2018. Even though some of them (eg Kerminen et al., 2018,
Lee et al 2019) provide insight on the parameters this study is also focusing on, none has gone been
as detailed as the one presented in this work. Therefore, the compilation of results presented in this
work are of interest to the community and would be worthwhile publishing if the manuscript was
well written and the analysis provided informative and concise. I am afraid that this is not the case.
After reading the article, I was disappointed not to find any information on seasonality for any of
the parameters investigated even though multi year data were investigated.
RESPONSE:  It is not clear by the comment what kind of analysis is expected (whether it is the
seasonality of the parameters themselves or the seasonality of their effect). The seasonality of the
parameters (which was found to favour mainly summer for the growth rate, while the results for the
formation rate were more variable) is separately investigated in a previously published paper for
the UK sites (Bousiotis et al., 2019) and for the rest of the sites in an already submitted manuscript
(Bousiotis et al., 2020) and thus was not discussed in the present study (this is noted in 2.1). The
seasonality of their effect was not studied in the present manuscript as this would extend its size
too much. It is worth pointing out that the variables most affected by season such as temperature
and insolation are considered in this paper, and to break down the data analysis according to
season would involve a great deal of repetition.

Furthermore the authors fail to deliver any error metric whatsoever (deviation, error, confidence
level). The lack of the most elementary statistical analysis was striking.
**RESPONSE:** Much of this information is included in the paper. Deviation errors are included in
the SI figures for every subgroup of every variable studied (reporting these is unrealistic as they are
over a thousand). $R^2$ is reported for every slope calculated of every variable studied on the figures.
p-values are reported (when significant) for every variable in every site in tables 3 and 4. We have
calculated, but not included, the error of the slope for every variable calculated using the normalised
gradients, but have not included this as the normalised slopes do not have any significance other
than their absolute value; we include only information on their trend.
The other striking feature is the poor use of English and terminology, which I explain thoroughly
below. The use of English must be improved as there are many sentences that require revision.
**RESPONSE:** Many changes in terminology and corrections were applied throughout the
manuscript to improve the level of English.
The major drawback is the generalizations and uncertain phrases used throughout the manuscript.
The authors should be concise and specific instead.
**RESPONSE:** The manuscript was updated in many cases to reduce uncertainty (whenever there
was enough confidence in the statements presented)
As an example in Line 69 (76) it is advised to name the places (exceptions) were NPF is hardly
observed. A nice review can be found by Lee et al., 2019 (section 4.8).
**RESPONSE:** Exceptions where NPF events are not observed and references were added.
Example 2 Line 270: A few sites presented a strong correlation, which in all cases were background
sites (either rural or urban). A few sites (which ones?) presented a strong correlation (nowhere in
the manuscript strong, medium weak is defined. The reader has no idea what the author is
discussing) which in all cases were background sites (either rural or urban; to the best of my
knowledge rural sites are considered as background sites. What do the authors mean?). I assume
that the authors are trying to point at urban kerbside sites with this sentence, yet I am not really sure
what they mean. And the paragraph continues The relation (which one?) found in most cases (how
many, percentage?) was positive (does this mean a positive slope? Where is it shown? In which
table or graph?) apart from two roadsides (improper terminology) and GREUB, though due to the
low (again low is not defined?) R2 these results cannot be used with confidence (and where do the
authors draw the confidence line?). The above lines are just an example of improper phrasing used
throughout the manuscript that make it very hard to follow. Similar examples can be found
throughout the manuscript. A major drawback of this work is that many trends/relationships
reported are not referred to any table or Figure and hence are hard to follow.
**RESPONSE:** References to the sites mentioned in each case as well as $R^2$ values were added
throughout the manuscript to improve readability. References for the results were added in the beginning of each section (to avoid repetition). Specific references for unusual trends were also
added for the figures in the SI (figure numbering in SI was overhauled). References for SI figures
for simple relationships were not added as they are covered by the slopes found in tables 3 and 4.
Strong, weak and other characterisations of the correlations are now accompanied with either the $R^2$
or a range of the $R^2$.

NPF probability sounds to me as if you are trying to predict the occurrence of nucleation events. Based on Line 218 (Equations are not numbered!) a more suited term would be NPF frequency.
**RESPONSE:** The term NPF frequency is used within the text for the frequency of the events without taking into account any grouping of the data (into groups of condition ranges e.g NPF probability for RH in the range 60 – 65%). To separate these the term probability was used instead. Equation numbers were added.

The authors should consider adding reference formation and growth rates from other studies in their figures for comparison. I understand that this is not always possible (especially for formation rates) but is for the other two parameters in question.
**RESPONSE:** Similar to a previous comment, such an analysis was done in other studies either already submitted or published (Bousiotis et al., 2020; 2019).

Table 2 should also include growth rates for this study.
**RESPONSE:** The parameters of NPF are already reported in previously submitted studies (Bousiotis et al., 2020; 2019). The frequency and formation rate are reported here because they are used in the calculation of the normalised slopes, which is not done for the growth rate (see the methodology). Nevertheless, the growth rate and the number of NPF events for each site were added in Table 2.

The authors fail to summarize the seasonality of the parameters they are exploring even though they are having multi year data. This is very disappointing.
**RESPONSE:** This comment has already been addressed.

The statement in Lines 45-46 (49 – 50) is not true. Please read Kerminen et al., 2018 for example. That work which explicitly states the opposite.
**RESPONSE:** The sentence was rephrased into "without always following" to state that exceptions exist as pointed later in the Introduction part.

I have noted another case (Lines 98-99) (109 – 110) in the manuscript where the authors focus on the exceptions (which always exist) rather than the rule giving a very distorted view to the reader.
**RESPONSE:** In the text it is stated that "the negative effect of CS is widely accepted" and follows mentioning the exceptions found in the literature as "cases were found". This does not imply that the exceptions are anything more than that and it is essential that they are mentioned.

The introduction is very poor on references.
**RESPONSE:** More references were added in the introduction and throughout the manuscript

Lines 107-111 (118 – 123) should be rephrased. I cannot make sense of it at all.
**RESPONSE:** This sentence was included to show that the NPF events considered in our study
were not driven by combustion products but by secondary formation. A parenthesis was added
though which makes the sentence easier to understand.

Lines 82-84 (94 – 96). Please mention that increasing temperatures also have a negative effect as
they increase the energy barrier the clusters have to overcome to become stable and grow in size.
**RESPONSE:** The comment has been added.

Kerminen, V. M., Chen, X., Vakkari, V., Petäjä, T., Kulmala, M. and Bianchi, F.: Atmo- spheric
new particle formation and growth: Review of field observations, Environ. Res. Lett., 13(10),
doi:10.1088/1748-9326, 2018.

Lee, S. H., Gordon, H., Yu, H., Lehtipalo, K., Haley, R., Li, Y. and Zhang, R.: New Particle
Formation in the Atmosphere: From Molecular Clusters to Global Climate, J. Geophys. Res.
Atmos., 124(13), 7098–7146, doi:10.1029/2018JD029356, 2019.

**Anonymous Referee #2**
General comments
The focus of this study is to investigate the effect of meteorological conditions and atmospheric
composition on the occurrence of new particle formation (NPF) events at 16 sites (rural, urban
background and roadside) located in 6 European countries. The results are based on more than 85
years of meteorological and atmospheric composition data. The authors are using a binned linear
regression to find correlations between parameters such as windspeed, temperature, pressure, or
solar radiation intensity, ozone or volatile organic compounds mixing ratios to name a few, and the
occurrence of NPF, particle growth and the formation rates. This is an interesting study and of
interest to the community, however the following comments should be addressed before publishing.
On many occasions the authors claim that certain variables are weakly or strongly correlated but do
not provide any numbers or figure to support these statements. Please provide references to the
exact figures in the manuscript or in the supplemental material (SM). This is an issue reappearing
throughout the manuscript. Following on that, Figure S1 in SM contains many figures and only one
caption. These figures are not marked with a number/letter. Please consider adding individual
numbering (or introduce letters) so the respective figures corresponding to individual sites when
being discussed in the manuscript could be easily found in SM.
**RESPONSE:** This is addressed in a later comment

It looks like the authors use terms frequency of NPF occurrence and NPF probability
interchangeably. NPF probability doesn't really fit here since you do not predict NPF events.
However, the NPF probability term is explained in the text and in the equation (line 191; also please
number equations). In results, however, the authors are using term frequency of NPF occurrence
(line 245). Please clarify, review the explanation in text and use the correct term throughout the
manuscript. I assume what you want to use is NPF frequency.
**RESPONSE:** This is explained later
I understand you identify the number of days with NPF according to the method by Dal Maso et al.
(2005) with additional certain criteria. It would be good to report the numbers of NPF events for
each site (and season?). Please explain what days with "relevant data" are when calculating the
frequency.
**RESPONSE:** Two additional papers analysing in detail the conditions of the NPF events at all the
sites were either published (for UK) or were submitted (for the rest of the sites). This is noted in
section 2.1. A figure has been added to the SI to show the seasonality.
If I understood correctly to calculate the frequency, you divide the number of days identified as
NPF event-days by all days that you have data available or "relevant(?)" data available? I am
curious how does the frequency changes when you use the number of all days with all data and not
only with "relevant data available"? It would be good to mention this number somewhere in the
manuscript or in the SM? Following on the above, please explain what is in e.g. line 191 "available
data" and "given group"?
**RESPONSE:** The term "relevant data" refers to all the data available at each site and are
considered in each analysis (and of course when those are available). At each site the data were
almost in their entirety available and the limitation was in most cases the SMPS data (its availability
for each site is reported in Table 1). The data is always considered only when they are available for
each variable studied (by the code used in the analysis, as they were calculated one by one), so no
hours of data with missing values were included. In other words, when e.g. temperature was studied
only the hours with both SMPS and temperature data were considered.
More detail on the site selection criteria would be helpful. Do these sites belong to a network? How
were these sites selected? In Line 120: the authors mention "geographical region and type of
environment". I suggest adding more description on the sites (e.g. in SM), their characteristic and
typical meteorological conditions, e.g. features they have in common/differences, number of NPF
studied and identified.
**RESPONSE:** A justification for the sites chosen is given in the Site description section. As
mentioned earlier, the analysis of the events, as well as the typical conditions for all the sites were
given in two separate earlier papers.

Having this extensive dataset, I encourage the authors to discuss variability e.g. seasonal, site to
site/regional. A figure where you plot frequency of NPF occurrence or number of NPF events for
each station in each season (e.g. bar plot?) on (y axis), for each site type (x axis) would be helpful.
Exploring e.g. seasonal variability would add value to the paper.
**RESPONSE:** This was already done in two earlier papers (Bousiotis et al., 2020; 2019).
Where there any limitations of the study? If yes, these should be discussed. Further, errors should
be included.
**RESPONSE:** The study is pretty straightforward, and the only limitation was the lack of data for
some variables at some sites e.g. $SO_2$ data was not available at all sites. A comment was added for
this limitation at the end of 2.1 Site Description and Data Availability section.
Are there general trends for these three site types? Maybe you could discuss these more. It would be
helpful to highlight (e.g. text in bold) data in Table S1 e.g. significant correlations.
**RESPONSE:** As explained in the response to the first comment, these are provided in other studies
(Bousiotis et al., 2020; 2019). Stronger correlations were highlighted with bold numbers.
What is the importance of the result of this study? The authors could discuss it more e.g. in
conclusion. I feel that is missing in the current version.
**RESPONSE:** The statement: "This study, apart from providing insights into the effect of a number
of variables on the occurrence and development of NPF events in atmospheric conditions across
Europe, also shows the differences that climatic, land use and atmospheric composition variations
cause to those effects. Such variations are probably the cause of the differences found among
previous studies." was added in the last paragraph of the text (838).
Please improve the language. It is critical to make the text more concise and clearer. It is hard to
follow the line of thoughts at points. There are some repetitions and long sentences that could be
shortened (e.g. lines: 76-82, 107-111, 325-330).
**RESPONSE:** Many changes were made in the manuscript to improve readability.
Specific comments
Line 38-62: in the abstract the authors could also mention: 85 years of data; how good these
correlations are (r2)' mention "meteorological conditions" e.g. such as …
**RESPONSE:** The information that a combined dataset of 85 years was used was added (52).
Added the highest $R^2$ values found for some variables (54 – 56). Added "(such as solar radiation
and relative humidity)" (58)
Line 42 (46): "except at very clean air sites" – more information is needed to this statement.
Something is missing. Please review or explain.
**RESPONSE:** This phrase has been removed.

Line 54 (60): What "higher values" means there exactly? Provide a number.
**RESPONSE:**  Added the word "average". No values can be provided as what is implied by the
results is that the importance of some variables becomes less as the average values (average
conditions) become higher or lower, depending on the general trend.
Line 61- 62 (67-68): you could give these values in brackets
**RESPONSE:**  No values were added as the text implies that one increases with the other
simultaneously, similar to the meteorological conditions.
Line 97 (108): "negative effect" on?
"on the occurrence of the events" was added.
Line 99 (110): "average conditions"? What does it mean here?
**RESPONSE:**  No change in the text. It means the average CS which is well covered with the term
"average conditions" in this case.
Line 107-111 (118 – 123): hard to follow, please review and shorten
**RESPONSE:**  Already mentioned by referee #1 and addressed.
Line 121 (133): please add references to the studies you refer to
**RESPONSE:**  References were added
Line 122 (134): NPF probability? Or NPF occurrence? As mentioned above, probability doesn't
really fit here
**RESPONSE:**  NPF probability was not changed as every time it is mentioned it implies the results
from the analysis/modelling that was done in this study. An explanation for this was provided (213)
Line 124 (138): I suggest calling this section: "2. Methods", 2.1 as is. 2.2 as is or similar. This way
you can remove 2.2 Methods so it does not appear twice. In 2.1 the authors could mention which
cities/countries/sites were used; which meteorological and atmospheric composition variables did
you use in this study already at this point. Which stations had a full set of data and which only some
etc. Maybe also mention which are dependent and which independent variables. And what do you
consider relevant data days, what do you mean by available data: e.g. in line 189. Please be more
specific upfront. You could also add information on how these sites were chosen? Any criteria you
applied to select these? Are they belong to a network? Are they similar or different in any respect?
**RESPONSE:**  Section naming was not changed as it is considered sensible for a chapter named
"Data and Methods" to have section 1 named "Sites and data" and section 2 as "Methods". The
countries and cities included in the study are mentioned. A list of the data available in each site is
found in Table 1, as mentioned in the text. A justification for the sites chosen was also added. The
sites do not belong to a single network and thus such information is not provided.

Line 127 (140): I feel that the number of events (1950) is already a result so it should go to the
result section and not methods. Also, it is mentioned before the description of the NPF selection
method itself.
**RESPONSE:** The number of events was moved to the beginning of the Results section
Line 131 (142): it is also referring to the result. I suggest moving this sentence to the result section.
**RESPONSE:** The reference for the Table with results was moved to the beginning of the Results
section.
Line 136-143 (156 – 165): please add more details to the approach taken in this study. What "Ia"
exactly refers to and which additional criteria was used (line 142).
**RESPONSE:** The process of NPF event extraction was rewritten and more details were added for
the approach taken (156 – 169).
Line 137 (158): add size range of the nucleation mode you consider in your study
**RESPONSE:** Added "(smaller than 20 nm in diameter)".
Line 139 (160): you could mention confidence level in the brackets
**RESPONSE:** Changed to level of certainty to avoid misunderstandings.
Line 151 (178): add "respectively" after "particles". You could already mention there Fuchs
correction factor and keep it explained below.
**RESPONSE:** Added the word "respectively" (178). Second was not mentioned to avoid repetition
and flow distraction.
Line 149 (176): Formulas need to be numbered
**RESPONSE:** Equation numbers were added
Line 188 (216): given group? please explain
**RESPONSE:** The NPF probability is calculated for the range of data in a specific group (time
range, range of a given variable ex. for relative humidity from 50 to 55% etc.). Text was slightly
modified to reflect this better.
Line 191 (219): Again: I am not sure I follow this equation: what are these groups? Is it just a
number of days with NPF that was accompany by all relevant data? From explanation you seem
only to take days with NPF that were accompanied by relevant data.
**RESPONSE:** For the analysis done, data was separated into smaller groups as mentioned earlier.
Thus, the term probability is considered more appropriate than frequency. This is clearly stated in
the text.

Line 196 (224): low significance? Please give a number
**RESPONSE:** The results found from the analysis of raw data, due to the large spread, almost never
provided with any significant result (the $R^2$ was always very low). A single number cannot be
provided as the results are numerous. The word "statistical" was added.
Line 212 (241): extreme values? Please give a number
**RESPONSE:** As previous. The cases that extreme values that biased the results were many. For
example, an extreme value of wind speed (a single very windy day with no event) would result in
an NPF probability of zero for that wind speed range. This though would result in biasing the whole
analysis by a very limited range of data.
Line 239 (268): Results and Discussion? You could include here sentence in line 126: You mention
1950 events studied, could you provide information on how many were identified? It would be
helpful if authors mention that in the paper a summary of data can be found in the manuscript and in
the SM data/results for individual sites is presented.
**RESPONSE:** The sentence mentioned (the number of events extracted) was moved to the Results
section as suggested. The events studied were those that all the work was focused on. While other,
less clear events (without the expected growth, advected, uncertain etc.) were also extracted, they
were considered only as exceptions or special cases in previous works. For further information
about the events for each site as well as the comparative study between them, references were added
in the Site Description section.
Line 245 (277): what is relevant data? Please explain more clearly in methods section and refer to it.
Diagnostic features – wouldn't these be better in methods?
**RESPONSE:** Added "depending on the variable studied". Relevant data refer to the data available
depending on the variables studied, e.g. to find the frequency of NPF events, the days with available
SMPS data were only considered. These diagnostic features are used to present the results in Table
2 and thus were not moved to Methods.
Line 252 (286): "slopes and R2" please use correct terms for these or more careful description
**RESPONSE:** Changed to the terms gradient (instead of slope) and coefficient of determination for
$R^2$.
Line 261 (296): very strong? Please provide references to the exact figures in the
figures/supplemental material when discussing results
**RESPONSE:** In this case very strong correlations were considered for $R^2 > 0.75$ as explained in the
parenthesis (and a clarification is added to any characterised correlation). The correlations (whether
weak or strong) are found in Tables 3 and 4 (references added). References to the figures in the SI

are not needed when discussing slopes and $R^2$ and were only referenced when variable/unusual
trends were found.
Line 279 (315): low? Please give a number and refer to the figure. Also, you placed all figures in
the SM under Figure 1. Maybe it would be better for the reader to have them split into different
figure numbers or a,b,c,d? This way it would be easier to find the one you describe at the very
moment.
**RESPONSE:**  The value of the $R^2$ was added in a parenthesis. Also, changed the numbering scheme
for the figures in SI. References to figures in the SI that present results not in the tables (i.e. variable
trends) were added in the text.
Line 296 (333): reference?
**RESPONSE:**  A reference was added
Line 301-303 (339): why? Could you explain? When describing results maybe worth mentioning
these for various site specifics? Anything in common?
**RESPONSE:**  A possible explanation was added "This may be due to the different seasonality of
the events found for the Greek sites (being more balanced within a year), as there was increased
probability of NPF events for the seasons with higher RH compared to other sites, making it a less
important factor for their occurrence."
Line 369 (414): which factors remain constant?
**RESPONSE:**  Added the word "meteorological"
Line 377 (420): reference?
**RESPONSE:**  References for this are found in the introduction
Line: 398 (441 - 443): maximum? Low?
**RESPONSE:**  Maximum changed to greatest. Low wind speeds changed to "close to zero wind
speeds".
Line 420 (464): Ethesian: add few words what these are could be added
**RESPONSE:**  A brief description has been added ("a pressure system that develops in the region
every summer").
Table 3: what is a "p value"? has it been defined somewhere? In tables: the authors could use bold
text to highlight significant correlations? So these patterns/trends could be clearly seen?
**RESPONSE:**  Added the definition of p-value (line 286). Used bold text for all correlation $r > 0.50$
Figures: no need to mention in each caption "of the present studies"

**RESPONSE:** The phrase was removed
Line 433 (479): you could already mention here which pollutants (such as. . .) are studied and
described in the upcoming sections.
Added the chemical compounds studied in a parenthesis
Line 752 (806): "at higher values"?
Changed to "at sites with higher average values"
Line 755 (810): "meteorological conditions" such as?
"such as temperature or relative humidity" was added
Line 756-757 (812): is that the only explanation? How about chemistry/composition at such type of
site? Anything else that might play a role?
**RESPONSE:** Added "compared to the urban environments and the more complex chemical
interactions found there"
Line 782-783 (840): seems out of place here; it would be more suitable at the beginning of
conclusion section or removed.
**RESPONSE:** Moved the sentence to the start of the Conclusions section (line 796).
Interactive comment on Atmos. Chem. Phys. Discuss., https://doi.org/10.5194/acp-2020-555, 2020
############################################################
**RESPONSE:** Additionally, four authors were added in the list of authors
Table 1 was updated

[revised manuscript text omitted]

(a)

[Figure]

(b)

[Figure]

**(c)**

**(d)**

[Figure]

**(e)**

**(f)**

[Figure]

**(g)**

[Figure]

**(h)**

[Figure]

**(i)**

[Figure]

**(j)**

[Figure]

[Figure]

[Figure]

**(m)**

[Figure]

**(n)**

[Figure]

**(o)**

[Figure]

**(p)**

[Figure]

                                                                                                                          **(q)**

[Figure]

                                                                                                                          **(r)**

[Figure]

**Figure S2:** Relationship of temperature with NPF variables.

[Figure]

(a)

[Figure]

(b)

[Figure]

**(c)**

**(d)**

[Figure]

(e)

(f)

[Figure]

**(g)**

**(h)**

[Figure]

(i)

(j)

[Figure]

**(k)**

**(l)**

[Figure]

**(m)**

**(n)**

[Figure]

[Figure]

**(q)**

**(r)**

**Figure S3:** Relationship of relative humidity with NPF variables.

[Figure]

**(a)**

[Figure]

**(b)**

[Figure]

(c)

[Figure]

(d)

[Figure]

[Figure]

[Figure]

**(g)**

[Figure]

**(h)**

[Figure]

(i)

[Figure]

(j)

[Figure]

**(k)**

[Figure]

**(l)**

[Figure]

                                                  **(m)**

[Figure]

                                                  **(n)**

[Figure]

**(o)**

[Figure]

**(p)**

[Figure]

**(q)**

**(r)**

**Figure S4:** Relationship of wind speed with NPF variables.

[Figure]

**(a)**

[Figure]

**(b)**

[Figure]

(c)

(d)

[Figure]

**(e)**

**(f)**

[Figure]

**(g)**

[Figure]

**(h)**

[Figure]

(i)

(j)

[Figure]

[Figure]

[Figure]

**(m)**

[Figure]

**(n)**

[Figure]

**(o)**

[Figure]

**(p)**

[Figure]

(q)

[Figure]

(r)

**Figure S5:** Relationship of atmospheric pressure with NPF variables.

[Figure]

(a)

[Figure]

(b)

[Figure]

**(c)**

[Figure]

**(d)**

[Figure]

(e)

[Figure]

(f)

[Figure]

(g)

[Figure]

**(h)**

**(i)**

[Figure]

(j)

[Figure]

[Figure]

[Figure]

                                                                                **(m)**

[Figure]

                                                                                **(n)**

[Figure]

(o)

**Figure S6:** Relationship of SO₂ concentration with NPF variables.

[Figure]

**(a)**

[Figure]

**(b)**

[Figure]

**(c)**

**(d)**

[Figure]

**(e)**

[Figure]

**(f)**

[Figure]

[Figure]

[Figure]

[Figure]

[Figure]

[Figure]

**Figure S7:** Relationship of NO$_2$ / NO$_x$ concentration with NPF variables.

[Figure]

**(a)**

[Figure]

**(b)**

**\*NO$_2$ for SPARU and GRERU**

[Figure]

**(c)**

**\*NO₂ for SPAUB**

[Figure]

**(d)**

[Figure]

(e)

[Figure]

(f)

*NO$_2$ for SPARU and GRERU

[Figure]

**(g)**

*NO₂ for SPAUB

[Figure]

**(h)**

[Figure]

**(i)**

**\*J$_{16}$ for UKRU**

[Figure]

**(j)**

**\*NO$_2$ for SPARU and GRERU**

[Figure]

**(k)**

**\*J₁₆ for UKUB**

**\*\* NO₂ for SPAUB**

[Figure]

**(l)**

**\*J₁₆ for UKRO**

**Figure S8:** Relationship of O$_3$ concentration with NPF variables.

[Figure]

**(a)**

[Figure]

**(b)**

[Figure]

(c)

[Figure]

(d)

[Figure]

(e)

[Figure]

**(f)**

**(g)**

[Figure]

**(h)**

[Figure]

**(i)**

[Figure]

y = 0.0368x + 1.9021
R² = 0.1699

GRERU

(j)

[Figure]

                                                                        **(k)**

[Figure]

                                                                        **(l)**

[Figure]

**(m)**

[Figure]

**(n)**

[Figure]

**Figure S9:** Relationship of particulate organic carbon concentration with NPF variables.

[Figure]

**(a)**

**(b)**

[Figure]

[Figure]

**(d)**

[Figure]

**(e)**

[Figure]

(f)

[Figure]

                                                                          (g)

                                                                          (h)

[Figure]

**Figure S10:** Relationship of SO$_4{}^{2-}$ concentration with NPF variables.

[Figure]

**(a)**

[Figure]

**(b)**

[Figure]

(c)

[Figure]

**(d)**

**(e)**

[Figure]

[Figure]

**(g)**

[Figure]

**(h)**

[Figure]

                            **(i)**

**Figure S11:** Relationship of gaseous ammonia concentration with NPF variables.

[Figure]

 (a)

[Figure]

 (b)

[Figure]

446 **Figure S12:** Relationship of the condensation sink with NPF variables.

(a)

[Figure]

(b)

[Figure]

**(c)**

[Figure]

**(d)**

[Figure]

                                                                                (e)

                                                                                (f)

[Figure]

(g)

(h)

[Figure]

[Figure]

[Figure]

**(k)**

[Figure]

**(l)**

[Figure]

(m)

[Figure]

(n)

[Figure]

[Figure]

[Figure]

**(p)**

**(q)**

**Figure S13:** Relationship of average temperature and normalised gradients $a_N^*$ for all but the Finnish
sites.

**Figure S14:** Seasonal variation of NPF events

[Figure]

[Figure]

[Figure]

[Figure]

[Figure]

[Figure]

[Figure]

[Figure]

[Figure]

[Figure]

[Figure]

[Figure]

[Figure]

[Figure]

[Figure]

[Figure]

[Figure]

[Figure]

[Figure]

[Figure]

[Figure]

[Figure]

[Figure]

[Figure]

[Figure]

[Figure]

[Figure]

[Figure]

[Figure]

[Figure]

[Figure]

[Figure]

[Figure]

[Figure]

[Figure]

[Figure]

[Figure]

[Figure]

[Figure]

[Figure]

[Figure]

[Figure]

[Figure]

[Figure]

[Figure]

[Figure]

[Figure]

[Figure]

[Figure]

[Figure]

[Figure]

[Figure]

[Figure]

[Figure]

[Figure]

[Figure]

[Figure]

[Figure]

[Figure]

[Figure]

[Figure]

*NO₂ for SPARU and GRERU

[Figure]

*$NO_2$ for SPAUB*

[Figure]

[Figure]

[Figure]

*NO₂ for SPARU and GRERU

[Figure]

*NO₂ for SPAUB

*$NO_2$ for SPAUB

[Figure]

[Figure]

**\*J$_{16}$ for UKRU**

**\*NO$_2$ for SPARU and GRERU**

[Figure]

*J$_{16}$ for UKUB
** NO$_2$ for SPAUB

[Figure]

*J$_{16}$ for UKRO

[Figure]

[Figure]

[Figure]

[Figure]

[Figure]

[Figure]

[Figure]

[Figure]

[Figure]

[Figure]

[Figure]

[Figure]

[Figure]

[Figure]

[Figure]

[Figure]

[Figure]

[Figure]

[Figure]

[Figure]

[Figure]

[Figure]

[Figure]

[Figure]

[Figure]

[Figure]

[Figure]

[Figure]

[Figure]

[Figure]

[Figure]

[Figure]

[Figure]

[Figure]

[Figure]

**Figure S2:** Relation of average temperature and normalised slopes $a_N^*$ for all but the Finnish sites.

---

## Author Response (AR3)

**MS No.:** acp-2020-555
**Title:** The Effect of Meteorological Conditions and Atmospheric Composition in the Occurrence and Development of New Particle Formation (NPF) Events in Europe
**Author(s):** Dimitrios Bousiotis et al.

**RESPONSE TO REVIEWERS**

**Referee #1**

General comments
The current version of the manuscript has improved. Readability is better. The authors followed most of the suggestions. However, there are still few minor aspects I encourage to improve.

The readability of the abstract and introduction could still be polished.
**RESPONSE:** We have done our best to improve these sections.

I am not convinced by the use of term "NPF probability". You are probably aware that some readers might only (or first) check the abstract and conclusion. Using "NPF probability" term may be confusing. If you still think this is something you would like to keep, I suggest be more specific what it actually represents when discussing it in conclusion.
**RESPONSE:** As the term "NPF probability" seems to cause a lot of misunderstandings it was replaced by the more conventional "NPF frequency" throughout the manuscript. Small changes were also made in the Methods section to accommodate the change (line 238).

Please specify how the NPF frequency was calculated, and check values presented in Table 2. Which data is considered for this calculation? I think the answer was given in author's response (AR) file but I missed mentioning of that in the main manuscript. Further, some comments are addressed in AR but these clarifications do not always appear in text. Please double check.
**RESPONSE:** Due to the aforementioned change (removal of the "NPF probability" term), the term "NPF frequency" is now explained in the Methods section with the inclusion of the "full dataset" as an option of the possible groups for which it is calculated (as the term "NPF frequency" was previously used when full datasets were considered). Additionally, some information about the trends and results included in the AR (pointing to previous work) were also added in the manuscript as they can help in explaining some trends (please see the response on a later comment as well).

How sure are you that NPF occurred at UK sites if you only have SMPS data available >16 nm for these locations? Please comment in the manuscript. It is also one of the limitations of the study that you could mention. What was your approach for analysis of these sites?
**RESPONSE:** For the extraction of the NPF events in the UK sites additional criteria were set (including the variation of particle number concentration data from 7 nm, the variations of pollutant concentrations, the condensation sink, the effect of the nearby Heathrow Airport, etc.). The text from the Methods section in the paper in which these results were first presented reads:
"At this point it should be mentioned that due to the particle size range available, NPF events in which new formed particles failed to grow beyond 16.6 nm (if any) could not be identified. Bursts of new particles in the size range < 16.6 nm that were identified using the CPC data but did not appear in the SMPS dataset were ignored as their development was unknown. This type of development was rare and mainly found at the rural background site, occurring on a few days per year mainly in summer. Its main feature was the short duration of the bursts compared to event days. In the urban sites, this type of development was almost non-existent. High time resolution data for gaseous pollutants and aerosol constituents was used to identify pollution events affecting particle concentrations and these were removed from the data analysis. This analysis took account of the fact that nanoparticle emissions from Heathrow Airport affect size distributions at London sites (Harrison et al., 2018), and such primary emission influences were not included as NPF events."

As this is a lengthy clarification, a note was added in the Methods section in the present study, mentioning the limitation and referencing the work where this is further explained:
 "As the available SMPS datasets for the sites in the U.K. are for particles of diameter greater than 16 nm, additional criteria were set to ensure the correct extraction of NPF events including the variations of the particle number concentrations from a Condensation Particle Counter (CPC – measuring from 7nm), as well as of the concentrations of gaseous pollutants and aerosol constituents (please refer to the Methods section in Bousiotis et al., 2019)." (line 186)

I have noticed that frequently two other papers published/submitted by Bousiotis et al (2019, 2020) are mentioned in AR. It would be good if the authors make sure that these are also mentioned in relevant places throughout the manuscript (mentioning the issue raised by reviewers e.g.: "xx was explored in Bousiotis et al. xxxx and is not the focus in this paper" or ""xx can be found in…").
Also make sure that crucial information on the study is provided in the current manuscript without the need to often look into two other papers to get a complete picture.
**RESPONSE:** We thank the reviewer for this suggestion.  Some references were added in the manuscript that point to results from the previous studies. Also, some crucial information found in these works were also added in several points in the manuscript, to clarify the points made (results about the seasonality of the GR and J, the variation of the temperature, CS etc.).

Table 3: please indicate in the caption what values "in bold" indicate
**RESPONSE:** A clarification for what the values in bold indicate was added in the captions of Tables 3, 4 and S1.

Figure 7a, 7b: make one *caption* for figure 7(a and b) and only indicate on corresponding plots "a" and "b".
**RESPONSE:** The figure was updated with a single caption. Also, the plot 4b was removed from the Figure Legends table as it was moved to the SI

**Referee #2**

This work compiles already published results from 16 sites located in six European cities. Within this huge task, the effort is to identify relationships between meteo variables, gas phase composition and aerosol organic content with key NPF variables such as NPF frequency (the authors name it NPF probability), growth rate and formation rate. Several findings within this work justify publication; two most striking is the nonlinear relation of temperature with NPF probability and the fact that increased solar irradiance reduces the probability for NPF.

Some issues require attention though.

Starting from most important to least severe

The authors have chosen to use only Ia events and as a result the probability shown in Table 2 is several factors smaller than those reported in literature for the same sites. However, from a brief search, the difference can be up to a factor of six. It is well understood why the authors made such a choice as formation and growth rates can be calculated reliably from these type of events only. How does this choice reflect to the results shown? I strongly support that for one site (e.g. Hyytiälä) an intercomparison is carried out to indicate to the reader the tentative differences. My recommendation is to do so only for NPF probability. This is critical as most studies in the end classify NPF as events, undefined and non-events lumping Ia, Ib and II classes into one. The authors should add to the caption of Table 1 the fact that only Ia events are considered.
**RESPONSE:** According to the results from the analysis of NPF events at the sites of the study it was found that the NPF events that did not meet the criteria for class Ia were up to double the number of those that are characterised as class Ia. Thus, in the methods section the following text was added:

"As only class Ia events were only considered, it is expected that the frequency of the events calculated should be lower than that expected if all types of events were included. This could result in values up to one third of those anticipated if all classes of events were considered. For the extent of this variation please refer to Bousiotis et al., (2019; 2020) in which there is an extended analysis of the NPF events for each site, including the special cases of NPF events that do not comply with the criteria set for class Ia." (line 316).

Additionally, the text was updated for the Table mentioning that the statistics refer to class Ia NPF events. (We assume that the caption of Table 2 is the one that needs updating).

How coarse is the time resolution of OC and sulfate measurements examined in this study? The first impression for the former is that they are derived from the thermal optical method. For the latter AMS is mentioned. However, AMS is typically used for short-term campaigns and this is a multi-year study. Do the authors mean ACSM? If the resolution of these measurements is coarser than 1 h, are they reliable to be used in NPF studies? The authors should clarify the time resolution of these measurements, both OC and SO4, in the manuscript and discuss any complications. If the authors indeed use AMS measurements what fraction of the time period discussed do they cover? Also, it would be worthwhile to mention the publications that refer to these measurements.
**RESPONSE:** ACSM data was used (updated in the text). The measurement resolution for all the sites for which such data was used is 1 hour. Data with 3-hour resolution or more was available but was not used as it would bias the results. The note "For all the sites, the data used in the present study are of either 1-hour resolution or less. Data with coarser resolution were omitted for reliability." was added in the Methods section for clarification (line 160). References for publications that reported the measurements of this study were added in the Site Description section.

In each pair of variables (e.g. NPF probability and RH; growth rate and temperature) presented in this work, there seems to be a norm and one (or more) site that is an exception. Since the authors cannot fully explain why (and this is perfectly understood), it is worthwhile to mention that in the abstract or the conclusions or in both.
**RESPONSE:** The note "though exceptions were found among the sites for all the variables studied" was added in the abstract (line 57). The note "in the majority of the sites (though exceptions were found as well, mostly in the southern sites)," was added in the Conclusions section to point out the exceptions found (line 863).

There is little relationship between RH and CS at most sites. Is this because CS was based on dried measurements and was not corrected for hygroscopic growth? This would be understood since chemical composition was lacking on most sites. Please discuss if CS was corrected for hygroscopic growth and how that affects the results presented.
**RESPONSE:** A note has been added in the methods that CS was not corrected for hygroscopic growth as well as for the effect this has on the results presented.

ANOVA is only valid for normally distributed populations. Have the authors tested for normality? The F-test is typically used. How did the authors treat skewed distributions? Please discuss.
**RESPONSE:** The Shapiro-Wilk test was used to assess the normality and the vast majority of the variables were found to have $p > 0.05$ and thus were considered as normal. This is probably due to the removal of the extreme values (for the calculations, 90% of each dataset was kept removing the extremely high and/or low values and the possible outliers included in them). While this was not done to promote the normality of the populations but to reduce the bias from extreme values, it indirectly assisted in making the distributions normal. For the few remaining (e.g. the growth rates associated with $SO_2$ concentrations for UKRO) for which normality was not present, the square root of the values of the variable were considered to achieve normality and proceed to the ANOVA test. This clarification was added in the Methods section (line 244).

In the supplement, several relationships are clearly non-linear, such as the temperature-NPF probability for a few sites, but the authors insist to use a linear fit (probably for consistency). May I ask the authors to note on the supplemental graphs in which cases linearity is not followed. The authors are better aware of the statistical significance of the related graphs than the reader is. In the case of Denmark Rural (S2b), it is not evident whether the deviation from linearity is statistically significant or not.
**RESPONSE:** As it is expected that most readers will not read the SI, such deviations are discussed in the text (one example is the case of the Danish rural site mentioned which is discussed in the text). A linear relationship is not always the best to describe the relationships found, but indeed was chosen for consistency (now discussed in the text – line 282). A metric for the consistency of the linearity can be given by the $R^2$ and the p-values (e.g. when $R^2$ is low then the linearity is not consistent, at least statistically). Apart from that, it is unknown even to the authors whether a trend that starts at the extreme values of a variable (e.g. the decline found in the Danish rural site with the NPF frequency at high temperatures) would consistently continue if the temperature increases further or it is an artefact, and thus it was decided not to be further discussed in detail, apart from the mentions made in the text (as only speculation can be made).

Please define in the methods section what is weak, strong and very strong correlation in this work. It will assist the reader further.
**RESPONSE:** As there is no specific mention for the relationships in the Methods section, the definition of weak, strong and very strong correlations is mentioned in the first reference to the coefficient of determination (line 334).

The effect of SO2 on NPF that the authors are discussing in Section 3.2.1 has been presented before. Please check the references below. These works relate particle acidity to NPF. Experimentally it has also been verified at the site named GRERU in this work.
**RESPONSE:** The works suggested are mentioned and referenced in the $SO_2$ section (line 548).

Line 247. "The remaining data" is better use of English than the "data left"
**RESPONSE:** Text changed to "remaining data" (now line 278).

If the authors prefer the term NPF probability it is fine. But please use it throughout the manuscript. The caption in Table 2 is a bit confusing.
**RESPONSE:** The "NPF probability" was addressed in an earlier comment. The caption in Table 2 was updated.

**2.2 Methods**

**2.2.1 NPF events selection**

[revised manuscript text omitted]

The NPF frequency was calculated by the number of NPF event days divided by the number of days with available data in the given group (full dataset or temporal, variable ranges etc.). The results presented in this study were normalised according to the data availability, as:

$$NPF_{\text{}frequency} = \frac{N_{NPF\ event\ days\ for\ group\ of\ days\ X}}{N_{days\ with\ available\ data\ for\ group\ of\ da\cdots}} \qquad (\underline{7}\text{}$$

Finally, the p-values reported in the analysis derive from the ANOVA one-way test. As the normality of the variables is required for such an analysis, the Shapiro-Wilk test was used to assess the normality and the vast majority of the variables were found to have $p > 0.05$ and thus were considered as normal. This is probably due to the removal of the extreme values (as mentioned in section 2.2.3, for the calculations 90% of each dataset was kept removing the extremely high and/or low values and the possible outliers included in them). While this was not done to promote the normality of the populations but to reduce the bias from extreme values, it indirectly assisted in making the distributions normal. For the few remaining (e.g. the growth rates associated with $SO_2$

concentrations for UKRO) for which normality was not present, the square root of the values of the variable were considered to achieve normality and proceed to the ANOVA test.

**2.2.3    Calculation of the gradient and intercept for the variables used**

[revised manuscript text omitted]

• the frequency of events occurring (i.e. days with an event divided by total days with relevant data, depending on the variable and range studied), As only class Ia events were only considered, it is expected that the frequency of the events calculated should be lower than the expected one if all types of events were included. This could result in values up to one third of those anticipated if all types of events were considered. For the extent of this variation please refer to Bousiotis et al., (2019; 2020) in which there is an extended analysis of the NPF events for each site, including the special cases of NPF events that do not comply for the criteria set for class Ia.

- the rate of particle formation at a given size ($J_{10}$ in this case), which was found to have unclear seasonal trends among the sites and was higher for urban sites compared to rural in most cases (Bousiotis, 2019; 2020)

- the growth rate of particles from the lower measurement limit to 30 nm (or 50 nm for the UK sites), which was found to be greater during summer months for most of the sites, also studied in the aforementioned works.

From the analysis of the extended dataset a total of 1952 NPF events were extracted and studied. The NPF frequency, growth and formation rate for each site is found in Table 2. The seasonal variation of NPF events is found in Figure S14.

**3.1 Meteorological Conditions**

The gradients, coefficients of determination ($R^2$ – the relationships found are characterised as weak for $R^2 < 0.50$, strong for $0.50 < R^2 < 0.75$ and very strong for $
[revised manuscript text omitted]

Jung, J., Adams P. J., and Pandis, S. N., Simulating the size distribution and chemical composition of ultrafine particles during nucleation events, Atmos. Environ., 40, 2248–2259, doi:10.1016/j.atmosenv.2005.09.082, 2006.

Jung, J. G., Pandis, S. N., and Adams, P. J., Evaluation of nucleation theories in a sulfur-rich environment, Aerosol Sci. Technol., 42, 495–504, doi:10.1080/02786820802187085, 2008.

Kalivitis, N., Kerminen, V-M., Kouvarakis, G., Stavroulas, I., Tzitzikalaki, E., Kalkavouras, P., Daskalakis, N., Myriokefalitakis, S., Bougatioti, A., Manninen, H. E., Roldin, P., Petäjä, T., Boy,

M., Kulmala, M., Kanakidou, M. and Mihalopoulos N.: Formation and growth of atmospheric nanoparticles in the eastern Mediterranean: Results from long-term measurements and process simulations', Atmospheric Chemistry and Physics Discussions, pp. 1–38. 
[revised manuscript text omitted]

von Bismarck-Osten, C., Birmili, W., Ketzel, M. and Weber, S.: Statistical modelling of aerosol particle number size distributions in urban and rural environments - A multi-site study, Urban Climate, 11(C), pp. 51–66. doi: 10.1016/j.uclim.2014.11.004, 2015.

von Bismarck-Osten, C. and Weber, S.: A uniform classification of aerosol signature size distributions based on regression-guided and observational cluster analysis, Atmospheric Environment, 89, pp. 346–357. doi: 10.1016/j.atmosenv.2014.02.050, 2014.

von Bismarck-Osten, C. Birmili, W., Ketzel, M., Massling, A., Petäjä, T. and Weber, S.: Characterization of parameters influencing the spatio-temporal variability of urban particle number size distributions in four European cities, Atmospheric Environment, 77, pp. 415–429. 
[revised manuscript text omitted]
^*$ for the sites with available data (a) and for the sites with available data excluding UKRO (b).

[Figure]

**Figure 8:** Relationship of average $O_3$ concentrations and normalised gradients $a_N^*$.